# NFIB facilitates replication licensing by acting as a genome organizer

Wenting Zhang [1,7], Yue Wang[1,2,7], Yongjie Liu[3,7], Cuifang Liu[4], Yizhou Wang[4], Lin He[1], Xiao Cheng[1], Yani Peng[1], Lu Xia[1], Xiaodi Wu[5], Jiajing Wu[5], Yu Zhang [1], Luyang Sun[1], Ping Chen [6], Guohong Li [4], Qiang Tu [3], Jing Liang [1] ✉ & Yongfeng Shang [1,2] ✉

The chromatin-based rule governing the selection and activation of replication origins in metazoans remains to be investigated. Here we report that NFIB, a member of Nuclear Factor I (NFI) family that was initially purified in host cells to promote adenoviral DNA replication but has since mainly been investigated in transcription regulation, is physically associated with the pre-replication complex (pre-RC) in mammalian cells. Genomic analyses reveal that NFIB facilitates the assembly of the pre-RC by increasing chromatin accessibility. Nucleosome binding and single-molecule magnetic tweezers shows that NFIB binds to and opens up nucleosomes. Transmission electron microscopy indicates that NFIB promotes nucleosome eviction on parental chromatin. NFIB deficiency leads to alterations of chromosome contacts/compartments in both $G_1$ and S phase and affects the firing of a subset of origins at early-replication domains. Significantly, cancer-associated NFIB overexpression provokes gene duplication and genomic alterations recapitulating the genetic aberrance in clinical breast cancer and empowering cancer cells to dynamically evolve growth advantage and drug resistance. Together, these results point a role for NFIB in facilitating replication licensing by acting as a genome organizer, shedding new lights on the biological function of NFIB and on the replication origin selection in eukaryotes.

DNA replication holds the essence of genetic inheritance, in which exquisite mechanisms are implemented to ensure the genetic material is duplicated once and only once during each cell cycle[1–3]. In eukaryotes, replication initiates at multiple origins that are "licensed" first and subsequently "fire" to activate DNA synthesis[4]. Licensing of replication origins takes place in $G_1$ phase, beginning with the binding of the origin recognition complex (ORC) to origins, followed by CDC6

(cell division cycle protein 6) and CDT1 (chromatin licensing and DNA replication factor 1)-mediated loading of the minichromosome maintenance (MCM) double hexamer complex to form the pre-replication complex (pre-RC)[5]. As the cell cycle progresses into S phase, origins "fire" and replication initiates upon the pre-RC being transformed into the pre-initiation complex (pre-IC) through the recruitment of TopBP1, Treslin, Cdc45, and GINS to the pre-RC[5–7]. Accurate duplication of the

[1]Department of Biochemistry and Molecular Biology, School of Basic Medical Sciences, Peking University Health Science Center, Beijing 100191, China. [2]Department of Biochemistry and Molecular Biology, School of Basic Medical Sciences, Hangzhou Normal University, Hangzhou 311121, China. [3]State Key Laboratory of Molecular Developmental Biology, Institute of Genetics and Developmental Biology, Chinese Academy of Sciences, University of Chinese Academy of Sciences, Beijing 100101, China. [4]National Laboratory of Biomacromolecules, Institute of Biophysics, Chinese Academy of Sciences, Beijing 100101, China. [5]Department of Biochemistry and Molecular Biology, School of Basic Medical Sciences, Capital Medical University, Beijing 100069, China. [6]Department of Immunology, School of Basic Medical Sciences, Advanced Innovation Center for Human Brain Protection, Capital Medical University, Beijing 100069, China. [7]These authors contributed equally: Wenting Zhang, Yue Wang, Yongjie Liu. ✉e-mail: liang_jing@hsc.pku.edu.cn; yshang@hsc.pku.edu.cn

mammalian genome relies on sequential activation of 30,000 to 50,000 origins distributed on the genome with an average interval of -100 kb[8]. Origin firing during S phase exhibits a temporal pattern: euchromatic origins generally fire early in S phase, whereas heterochromatic origins fire later[3,9,10].

For decades, the precise genomic locations of replication origins have been under intensive investigation. Despite the early success in mapping origins with specific DNA sequences in *Saccharomyces cerevisiae*[11], no consensus signature or signatures predictive of replication origins have been identified in metazoan genomes. Meanwhile, it is becoming increasingly clear that ORC binding in metazoan genomes is largely independent of a specific DNA sequence but is highly influenced by local chromatin configuration[12–16]. In addition, it appears that ORC binding alone is not sufficient for the pre-RC assembly; other factors such as chromatin modifiers, histone chaperones, and histones themselves that impact chromatin architecture also contribute to shaping the genomic landscape of replication initiation[15,17,18]. Moreover, intriguingly, binding of metazoan ORC proteins on chromatin is not sufficient to fire an origin[19], and the actual initiation of DNA replication is mediated by the replicative MCM helicase, which determines the frequency and timing of initiation events[20,21]. Furthermore, it is believed that only a subset of replication origins are used to replicate the eukaryotic genome at each cell cycle[22]; inactive or dormant origins are potential origins that are rarely used under normal conditions but can be activated in specific cellular programs or under certain cellular conditions[23]. Clearly, more investigations are needed to understand the chromatin-based rule that governs the selection and activation of metazoan origins.

The Nuclear Factor I (NFI) family consists of 4 members, NFIA, NFIB, NFIC, and NFIX, which are believed to interact with DNA as a homo- or hetero-dimer[24]. Surprisingly, despite that NFI was originally isolated from crude nuclear extracts of HeLa cells to stimulate the initiation of adenoviral DNA replication in vitro[25] and subsequently demonstrated to bind to GCCAAT and stimulate eukaryotic transcription as well as replication in a cell-free system[26], the major effort concerning the biological function of NFI proteins has been focusing on how these proteins regulate gene transcription in different biological contexts ranging from stem cell differentiation to the development of various cancers[27,28]. Among the NFI proteins, NFIB is ubiquitously expressed in human tissues[29,30] and frequently overexpressed/amplified in various types of cancer, such as small cell lung cancer, melanoma, and ER⁻ breast cancer[31–35]. Importantly, recurrent mutations and translocations of *NFIB* have also been reported in multiple types of cancer[28]. Interestingly, it was recently reported that NFIB promotes metastasis of small cell lung cancer (SCLC) through a widespread increase in chromatin accessibility to regulate a diverse of gene pathways[36]. Intriguingly, *NFIB* is the only NFI member that has been defined among the "cancer-related genes" in the Human Protein Atlas (https://www.proteinatlas.org). Clearly, the mechanistic action and pathophysiological function of NFIB need to be further investigated.

In this study, we report that NFIB is physically associated with the pre-RC and functionally linked to replication licensing. We demonstrate that NFIB promotes chromatin accessibility to facilitate the pre-RC assembly. We show that NFIB directly binds to nucleosomes and facilitates the eviction of parental histones. We show that cancer-associated NFIB overexpression elicits gene duplication and genetic alteration that mimic the genetic aberrance in clinical breast carcinomas. We explore the clinicopathological significance of NFIB overexpression-evoked genomic aberrance in breast carcinogenesis.

## Results

### NFIB is physically associated with the pre-replication complex on chromatin

As stated above, NFIB is frequently overexpressed/amplified in various cancers[31–35]. As molecular mechanisms underlying breast carcinogenesis

have been a long focus in our lab[37–42], to gain more mechanistic insights into the role of NFIB in carcinogenesis, we first performed a comparative analysis of public datasets to profile the expression pattern of the NFI family members. Querying the expression profile of NFI proteins in the Human Protein Atlas revealed a ubiquitous expression pattern of all the NFI members in various human tissues (https://www.proteinatlas.org) (Fig. 1a). We further collected a panel of human cancer cell lines derived from different tissues and performed real-time reverse transcriptase PCR (qPCR) analysis. We found that while NFIA, NFIB, NFIC, and NFIX are all expressed in these cells, the relative abundance of NFIB was generally higher than that of the other NFI members (Fig. 1b). Western blotting analysis with polyclonal antibodies against NFIB also confirmed that NFIB is expressed in all of these cell lines (Fig. 1b).

Next, we utilized an epitope-based proteomic screening with combined immunopurification and mass spectrometry to interrogate the NFIB interactome in vivo. In these experiments, FLAG-tagged NFIB was stably expressed in human osteosarcoma epithelial cell line U2OS, which is relatively easy to be synchronized. Cellular extracts were prepared and subjected to affinity purification using anti-FLAG affinity gels. After extensive washing, the bound proteins were eluted with excess FLAG peptides, resolved, and visualized by silver staining on SDS-PAGE (Fig. 1c). The protein bands were retrieved and analyzed by mass spectrometry. NFIB was co-purified with a number of proteins that belong to the DNA replication machinery, including MCM3, MCM4, MCM5, and MCM6, the components of the MCM complex, and CDT1, a key subunit of the pre-RC. Additional protein species, including SPT16 and SSRP1 (the constituents of FACT), RPA1, MYH9, and LRPPRC, as well as NFIC and NFIX, were also co-purified with NFIB. The detailed results of the mass spectrometric analysis are provided in Supplementary Table 1.

To verify the physical interaction of NFIB with the components of the DNA replication machinery, total proteins from U2OS cells were extracted and co-immunoprecipitation was performed with antibodies detecting the endogenous proteins. Immunoprecipitation (IP) with antibodies against NFIB followed by immunoblotting (IB) with antibodies against CDT1, MCM2, MCM3, MCM4, MCM5, MCM6, or MCM7 demonstrated that all these proteins were efficiently co-immunoprecipitated with NFIB (Fig. 1d, left). As stated above, both CDT1 and the MCM proteins are the components of the pre-RC[5]. Thus, these observations suggest that NFIB is associated with the pre-RC in vivo. Indeed, IP with antibodies against NFIB followed by IB with antibodies against ORC1 or CDC6, two other components of the pre-RC, showed that both ORC1 and CDC6 were also efficiently co-immunoprecipitated with NFIB (Fig. 1d, left).

The MCM complex exists in both soluble and chromatin fractions in cells, and only chromatin-associated MCM helicase functions in DNA replication[43]. To ask the question of whether or not the interaction between NFIB and the pre-RC is relevant to DNA replication, U2OS cells were synchronized at $G_1$/S boundary by double-thymidine block prior to collecting cell lysates and extracting chromatin fractions for co-immunoprecipitation assays. IP with antibodies against NFIB followed by IB with antibodies against ORC1, CDC6, CDT1, or MCM2-7 detected the physical association of NFIB with the pre-RC components in chromatin fraction (Fig. 1d, middle), supporting a notion that NFIB is functionally linked to the pre-RC on chromatin. Reciprocally, IP with antibodies against CDT1, MCM2, MCM3, MCM4, MCM5, MCM6, or MCM7 and IB with antibodies against NFIB with chromatin fraction from U2OS cells also showed that NFIB was efficiently co-immunoprecipitated with the components of the pre-RC (Fig. 1d, right). Collectively, the above results indicate that NFIB is physically associated with and functionally linked to the pre-RC in vivo.

To further support the physical association of NFIB with the pre-RC and to understand the molecular interaction involved in the association, glutathione S-transferase (GST) pull-down assays were performed using GST-fused NFIB and in vitro transcribed/translated

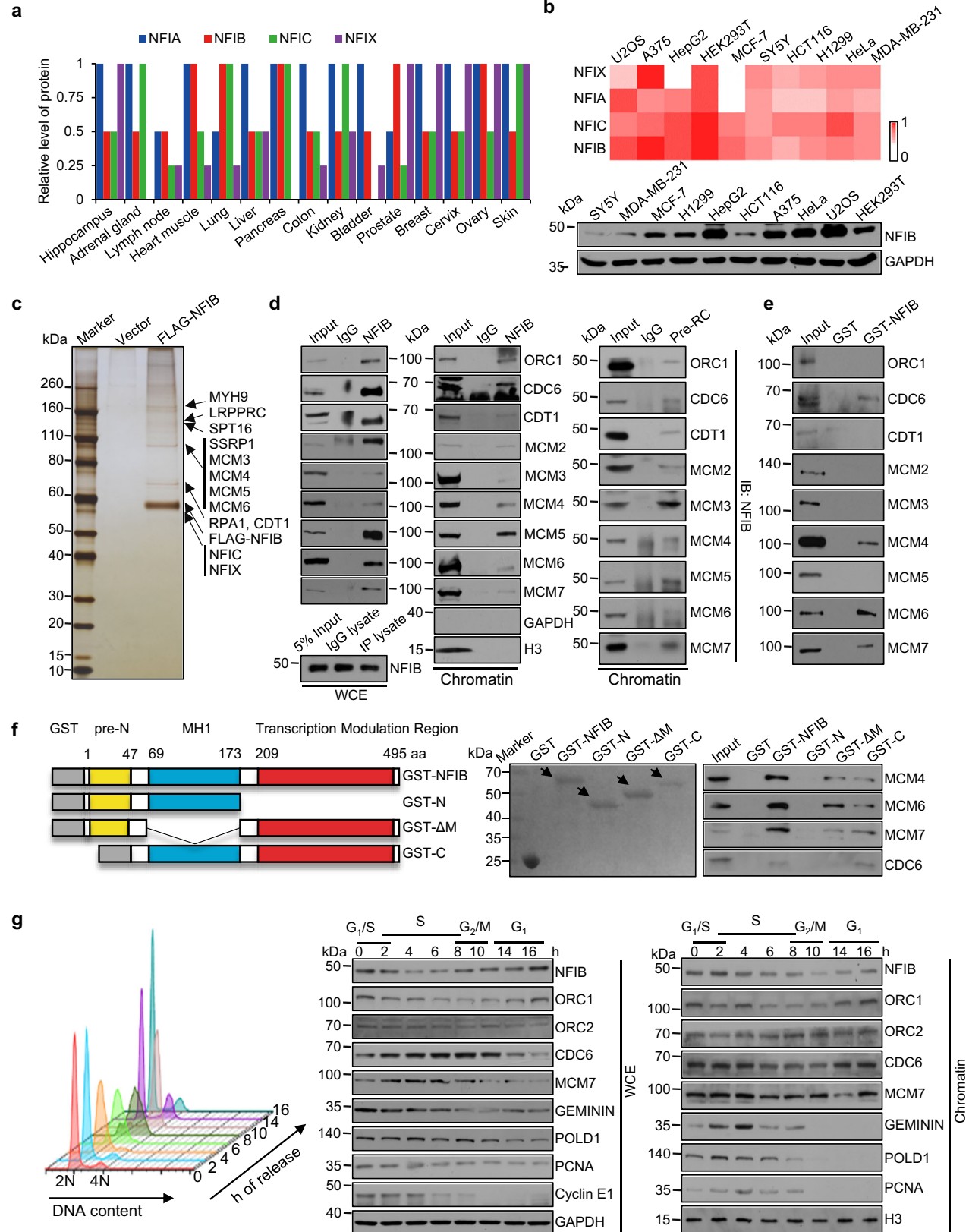

components of the pre-RC, including ORC1, CDC6, CDT1, and MCM2-7. The results showed that NFIB was capable of interacting with CDC6, as well as with MCM4, MCM6, and MCM7 (Fig. 1e), subunits that constitute the catalytic core of the MCM helicase[44], but not with the other pre-RC components that we tested. NFIB is mainly composed of two distinct structural modules: an N-terminal highly conserved DNA-

binding/dimerization domain (MH1) and a C-terminal transcription modulation region, plus a conserved pre-N-terminus region (8–47 aa) with unknown function[45] (Fig. 1f). Molecular interface mapping with GST-fused N-terminal fragment (1–173 aa, GST-N), MH1-deleted mutant (1–68 aa + 174–495 aa, GST-ΔM), or C-terminal fragment (69–495 aa, GST-C) of NFIB and in vitro transcribed/translated CDC6, MCM4,

**Fig. 1 | NFIB is physically associated with components of the Pre-RC. a** The relative protein level of NFI family members in different tissues from the Human Protein Atlas. **b** Analysis of the level of mRNA (upper) or protein (lower) of the NFI family members in different human cancer cell lines, by qPCR and western blotting, respectively. The heatmap was generated by log transformation of the data based on −ΔCT values. **c** Immunopurification and mass spectrometric analysis of NFIB-associated proteins. Cellular extracts from U2OS cells stably expressing FLAG-NFIB were affinity-purified. The eluates were resolved by SDS-PAGE and silver-stained. The protein bands were retrieved and analyzed by mass spectrometry. The experiment was performed twice with similar observations. **d** U2OS cells were synchronized at $G_1$/S boundary by a double-thymidine block. Co-immunoprecipitation assays were performed with whole cell extracts (WCE, left). The level of NFIB was examined by western blot (bottom). Co-immunoprecipitation assays were performed with chromatin fraction using antibodies against NFIB

followed by immunoblotting (IB) with antibodies against the indicated proteins (Chromatin, middle), or with antibodies against the indicated proteins followed by IB with anti-NFIB (Chromatin, right). **e** GST pull-down assays with GST-fused full-length NFIB and in vitro transcribed/translated proteins as indicated. **f** GST pull-down assays with deletion mutants of NFIB and in vitro transcribed/translated pre-RC components as indicated. Schematic diagrams of NFIB and NFIB deletion mutants are shown. Coomassie brilliant blue staining of GST-fused proteins is marked (arrow). **g** U2OS cells were synchronized at $G_1$/S boundary by double-thymidine block and released for different hours to allow cells to enter different phases of the cell cycle as indicated for cell cycle analysis by cytometry (left) or for western blotting analysis of whole cell extracts (WCE) or chromatin fraction (Chromatin) with antibodies against the indicated proteins. GAPDH or H3 was used as a loading control as indicated. (**b**, **d**–**g**) The experiment was performed three times with similar observations.

MCM6, or MCM7 revealed that the so-called transcription modulation region spanning 174–495 aa was responsible for the interaction of NFIB with MCM4/6/7, while the fragment covering 69–173 aa was responsible for the interaction of NFIB with CDC6 (Fig. 1f). These results reinforce the observation that NFIB is physically associated with the pre-RC in vivo.

To gain further support of the functional link between NFIB and the pre-RC and to explore the potential role of NFIB in DNA replication, we first examined the level of total cellular NFIB as well as chromatin-associated NFIB throughout cell cycle, along with the measurement of cell cycle-dependent expression and chromatin association of representative replication factors. To this end, U2OS cells were synchronized at $G_1$/S boundary by double-thymidine block followed by release for different times allowing cells to enter different phases of the cell cycle (Fig. 1g, left). Western blotting analysis of total cell lysates and chromatin-associated proteins showed that while total cellular NFIB accumulated in $G_1$ phase and gradually decreased as cells progressed to S phase (Fig. 1g, middle), chromatin-associated NFIB was abundant in $G_1$ phase and remained so in the early S phase (Fig. 1g, right). Remarkably, a similar spatiotemporal pattern of expression/distribution was also detected for the majority of the components of the pre-RC (Fig. 1g), strongly supporting a functional link between NFIB and the pre-RC.

## NFIB facilitates the Pre-RC assembly by increasing chromatin accessibility

To examine how NFIB could affect the assembly and function of the pre-RC, we then stably depleted NFIB in U2OS cells by lentivirally delivered NFIB shRNA. Total cell lysates and chromatin fractions were prepared from cells synchronized at $G_1$ phase, $G_1$/S boundary, or S phase. Western blotting revealed that depletion of NFIB did not affect the overall cellular levels of the pre-RC components, including ORC1, CDC6, and MCM7, in these phases of the cell cycle (Fig. 2a). However, NFIB deficiency was associated with a marked decrease in chromatin-bound ORC1 and CDC6 in $G_1$ cells and a greatly reduced chromatin-bound MCM7 in $G_1$/S cells, whereas it had limited effects on the chromatin binding of these factors in S phase cells (Fig. 2a). Remarkably, reconstitution of NFIB expression with a NFIB shRNA-resistant NFIB expression construct in NFIB-depleted cells could rescue the reduction of the chromatin loading of ORC1, CDC6, and MCM7 in $G_1$ and $G_1$/S U2OS cells (Fig. 2a). Notably, while chromatin-bound CDT1 also decreased in NFIB-depleted $G_1$ and $G_1$/S cells as the other components of the pre-RC did, the overall CDT1 protein level slightly increased upon NFIB knockdown (Fig. 2a), possibly due to some kind of compensatory mechanisms.

Immunofluorescent staining was performed next to further examine the effect of NFIB depletion on the loading of the pre-RC components onto chromatin. In these experiments, U2OS cells were pre-extracted with a nonionic detergent to remove soluble fractions prior to staining so that immunofluorescent signals mainly reflect

proteins that are associated with chromatin[46]. Immunofluorescent staining of pre-extracted U2OS cells showed that NFIB knockdown resulted in greatly reduced chromatin associations of ORC1, CDC6, CDT1, and MCM7 in $G_1$ and $G_1$/S cells, whereas the chromatin binding of ORC2 was not affected (Fig. 2b), consistent with a previous report that ORC2 constantly binds to chromatin in a cell cycle-independent manner[19]. On the other hand, immunofluorescent staining of un-extracted cells with antibodies against the replication factors showed that the overall signals for ORC1, ORC2, CDC6, CDT1, and MCM7 were largely unaffected by NFIB depletion (Supplementary Fig. 1a). Together, these results support a notion that NFIB is required for efficient chromatin loading of the pre-RC in $G_1$ phase of the cell cycle.

To further support the functional association between NFIB and the pre-RC loading, we next analyzed the genome-wide binding profile of NFIB and ORC1, a major component of the pre-RC that binds chromatin at an early stage, by cleavage under targets and tagmentation (CUT&Tag)[47]. In these experiments, U2OS cells were synchronized at $G_1$ phase, and NFIB- or ORC1- associated chromatin were immunoprecipitated and DNAs were amplified using non-biased conditions, labeled, and sequenced using Novaseq. Using MACS2 software with a $q$ value cutoff of 0.05, 27,579 NFIB-specific binding peaks and 22,915 ORC1-specific binding peaks were called. Cross-analysis yielded a total of 14,400 overlapping peaks, covering 52% of NFIB peaks and 62% of ORC1 peaks (Fig. 2c). In comparison, cross-analysis of ORC1 peaks with randomly sampled 27,579 genomic regions ten times generated only 0.64% (median) overlapping, demonstrating the significance of the co-binding between NFIB and ORC1. In addition, CUT&Tag was also performed in NFIB-depleted U2OS cells that were synchronized at $G_1$ to examine how NFIB might affect the genomic landscape of ORC1. Using a fold change cutoff of 1.2, we detected a total of 11,306 NFIB peaks and a sum of 5724 ORC1 peaks in overlapped regions with a decreased signal intensity upon knockdown of NFIB, compared to the peaks of NFIB and ORC1 called in U2OS cells without NFIB depletion. Eighty three percent (4752/5724) of the diminished ORC1 peaks coincided with a decreased NFIB signal intensity (Fig. 2c and Supplementary Fig. 1b), supporting a notion that NFIB is required for ORC1 binding at these regions, and they were then defined as N1 peaks for following cross-analysis.

We next performed ChIP-seq in U2OS cells synchronized at $G_1$/S with antibodies against H3K4me3 or H3K9me3, representing euchromatin or heterochromatin, respectively, and intercrossed the results with N1 peaks to understand the genomic distribution of NFIB-affected ORC1 binding. A total of 25,000 H3K4me3 peaks and 52,090 H3K9me3 peaks were detected in these experiments. The results clearly showed that the significantly enriched H3K4me3 signal but not H3K9me3 signal surrounding the N1 peaks (Fig. 2d), suggesting that NFIB mainly affects the pre-RC loading at euchromatin. To further investigate the genome-wide chromatin dynamics associated with NFIB-promoted pre-RC assembly, transposase-accessible chromatin combined with high-throughput sequencing (ATAC-seq) was then

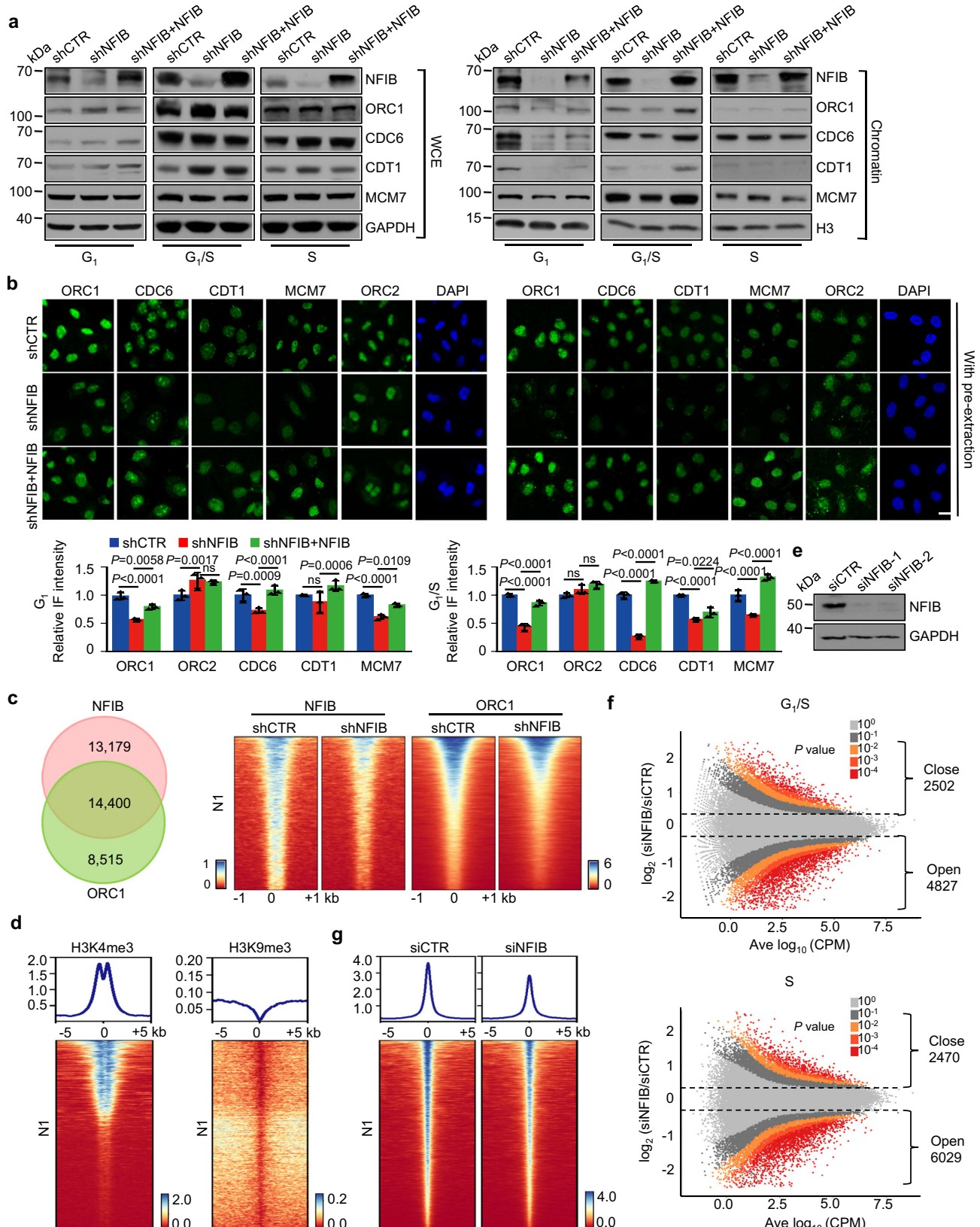

performed in U2OS cells with or without NFIB depletion. While NFIB-promoted the pre-RC assembly occurs mainly in $G_1$ and/or $G_1$/S phase as we showed earlier, NFIB-associated chromatin opening was also examined in S phase to reflect the replicative chromatin. To this end, U2OS cells treated with control siRNAs or siRNAs against NFIB were synchronized at $G_1$/S transition by double-thymidine block (Fig. 2e).

These cells were released for 0 or 4 h, representing $G_1$/S boundary or S phase, respectively, collected, lysed, and immediately subjected to transposition reaction followed by PCR amplification to add adaptor sequences for paired-end deep sequencing using the Illumina HiSeq system. The experiments were biologically repeated twice. MACS2 software was used for peak calling with a $q$ value cutoff of 0.05.

**Fig. 2 | NFIB facilitates the Pre-RC assembly by increasing chromatin accessibility. a** U2OS cells were infected with lentiviruses carrying control shRNA, NFIB shRNA, and/or FLAG-NFIB and synchronized at $G_1$ phase, $G_1$/S boundary or S phase. Whole cell extracts (WCE, left) or chromatin fraction (Chromatin, right) were prepared and analyzed by western blotting with antibodies against the indicated proteins. GAPDH or H3 was used as loading control as indicated. The experiment was performed three times with similar observations. **b** U2OS cells were infected with lentiviruses carrying control shRNA, NFIB shRNA, and/or FLAG-NFIB and synchronized at $G_1$ phase or $G_1$/S boundary for immunofluorescent staining (IF) using the indicated antibodies with pre-extraction. DAPI staining was included to visualize the nucleus (blue). Scale bar, 10 μm. Representative images from triplicate experiments are shown. The relative IF intensity was quantified by ZEN software. Data were presented as mean ± SD of ten sections from each slice for triplicate experiments. *P* values were determined by two-way ANOVA followed by Tukey test. NS, not significant. **c** U2OS cells were synchronized at $G_1$ phase for CUT&Tag experiments. Venn diagrams of the overlapping regions of NFIB binding and ORC1 binding identified by CUT&Tag of NFIB and ORC1 (left). Heatmaps showed the CUT&Tag signal distribution of 4752 N1 peaks (peaks with a decreased signal intensity of both NFIB and ORC1 in NFIB-depleted cells) (right). **d** Heatmaps showing the H3K4me3 and H3K9me3 signal distribution surrounding N1 peaks in U2OS cells synchronized at $G_1$/S phase. **e** The efficiency of knockdown was verified by western blotting in U2OS cells. The experiment was performed three times with similar observations. **f** U2OS cells without (siCTR) or with NFIB depletion (siNFIB) were synchronized at $G_1$/S or released into S phase for ATAC-seq. Differential accessibility between siCTR and siNFIB U2OS was plotted against the mean reads per region. Values aside brackets indicate the number of significantly changed peaks (*P* value < 0.01). **g** Heatmaps showing the distribution of chromatin accessibility signal in $G_1$/S phase surrounding N1 peaks in U2OS cells without (siCTR) or with NFIB depletion (siNFIB).

Deseq2 was used for differential peak analysis with *P* value cutoff of 0.01. Among the differential peaks identified in $G_1$/S cells, 4827 exhibited a decreased accessibility upon NFIB depletion, implying that NFIB is required for opening chromatin in these regions and thus were grouped as "open", whereas 2502 peaks showed increased accessibility upon NFIB depletion and thus were accordingly grouped as "close". NFIB-associated chromatin opening was more evident in S phase cells as revealed by 6029 peaks in the "open" group compared to 2470 peaks in the "close" group (Fig. 2f). Since NFIB promoted the pre-RC loading mainly at $G_1$ and/or $G_1$/S, we considered that the more evident NFIB-associated chromatin opening in S phase is likely due to secondary or amplification effect caused by genome-wide DNA replication. We then sought to examine the relationship between NFIB-associated ORC1 loading and chromatin openness. For this purpose, N1 peaks were cross-analyzed with the "open" group in $G_1$/S, which we considered more directly relevant. The results revealed that decreased ORC1 binding was indeed associated with decreased chromatin accessibility in NFIB-deficient cells (Fig. 2g), supporting a role for NFIB in opening up the chromatin to promote the pre-RC assembly.

## Depletion of NFIB alters replication profile and chromosome contacts/compartments

To investigate how NFIB-promoted the pre-RC assembly affects replication initiation, we next performed nascent strand sequencing (NS-seq) to identify *de facto* activated replication origins. To this end, RNA-primed nascent DNA at replication origins (0.5–2.0 kb in length) was purified from U2OS cells with or without NFIB depletion and subsequently sequenced using the Illumina platform[48]. Using MACS2 software with an FDR cutoff of 0.05 and after normalization to nascent-strand signals in cells treated with RNase A, a total of 50,986 NS-seq peaks were identified upon NFIB knockdown, of which 14,618 exhibited a decreased average signal intensity (S1, fold change > 1.2), 24,410 displayed an increased average signal intensity (S2, fold change > 1.2) and 11,958 were considered without a change in average signal intensity (S3, fold change < 1.2) (Fig. 3a, left), indicating that depletion of NFIB caused more of disordered replication pattern rather than decreased overall firing, likely because ceasing one origin is often compensated by igniting an alternate one. We also performed cross-analysis of ORC1 CUT&Tag, ATAC-seq, and NS-seq in wild-type U2OS cells to compare the ORC1 distribution with open chromatin and active origins. The results showed that 87.2% of ORC1 peaks (19,997/22,915) co-localized with ATAC-seq peaks, among which 21.6% (4320/19,997) co-localized with NS-seq-defined active origins (Supplementary Fig. 1c). The overlapping ratio is in agreement with previously reports containing similar assays[49–51]. Notably, the average signal intensity of NS-seq surrounding N1 peaks decreased as expected (Fig. 3a, right), supporting that NFIB-promoted ORC1 binding facilitates replication initiation at these regions. To further clarify whether NFIB-regulated origins have any preferred replication timing, we also performed Repli-seq in cells synchronized at early, middle, and late stages of S phase, respectively. For this purpose, U2OS cells with or without NFIB depletion were synchronized to $G_1$/S boundary by thymidine/aphidicolin double treatments prior to release for 0.5 h, 4 h, and 6 h, respectively. We found that NFIB-associated active origins (S1 peaks overlapped with N1 peaks, S1.N1) were more enriched at early-replication domains than other S1 peaks (S1.non-N1) or S2 peaks, whereas the overall NS-seq signals distributed widely throughout replication domains (Fig. 3b). While NFIB directly facilitates origin activity (S1.N1) at early-replication time, replication origins indirectly affected by NFIB (S1.non N1 or S2) preferentially locate at early-replication domains, conceivably because dormant origins fired by a compensatory mechanism are likely to close to each other. Notably, FACS analysis of the cell cycle progression U2OS cells indicated that while depletion of NFIB significantly inhibited S phase progression, the proportion of cells in early vs middle or late S-phases had no significant change (Supplementary Fig. 2), suggesting the preference of NFIB-affected replication origins were not an indirect effect due to altered proportion of cells within S phase. Representative Repli-seq tracks are shown in Fig. 3c.

To gain further understanding of the role of NFIB in influencing chromatin structure and DNA replication, we next performed Micro-C experiments to investigate the effect of NFIB on higher-order chromatin structure associated with DNA replication. U2OS cells infected with lentiviruses carrying control shRNA or NFIB shRNA were synchronized at $G_1$/S boundary by double-thymidine block or released for 4 h to allow cells to enter S phase. Significance analysis was performed with *t*-test of two replicates of each sample group in both $G_1$/S and S phase (*p* < 0.05). Upon depletion of NFIB, evident alteration of chromatin interaction was observed in both groups, with more significant changes seen in S phase cells (Fig. 3d). Similar to what we observed in the ATAC-seq, we considered that the more evident changes in S phase were due to amplification and/or indirect effects linked to global chromatin reconstitution during replication. Notably, more regions with weakened A compartment were identified in both $G_1$/S and S phase cells compared to other altered chromatin interactions, supporting that NFIB is associated with chromatin opening mainly at euchromatin (Fig. 3e). Representative micro-C maps are shown in Fig. 3f. We also calculated the number of NFIB-associated ORC1 binding (N1 peaks) within a 250-kb region of the altered compartments in the $G_1$/S phase. The results showed that N1 peaks were significantly enriched in the chromatin region featured with weakened A compartment but not regions with other features (Fig. 3g), supporting that NFIB-associated pre-RC loading is linked to increased accessibility at open chromatin.

To substantiate NFIB-promoted assembly of the pre-RC and the role of NFIB in DNA replication, we next examined the coexistence of NFIB and the pre-RC on two representative NFIB-associated replication origins, *ORI1* and *ORI2* (Fig. 4a). Quantitative chromatin

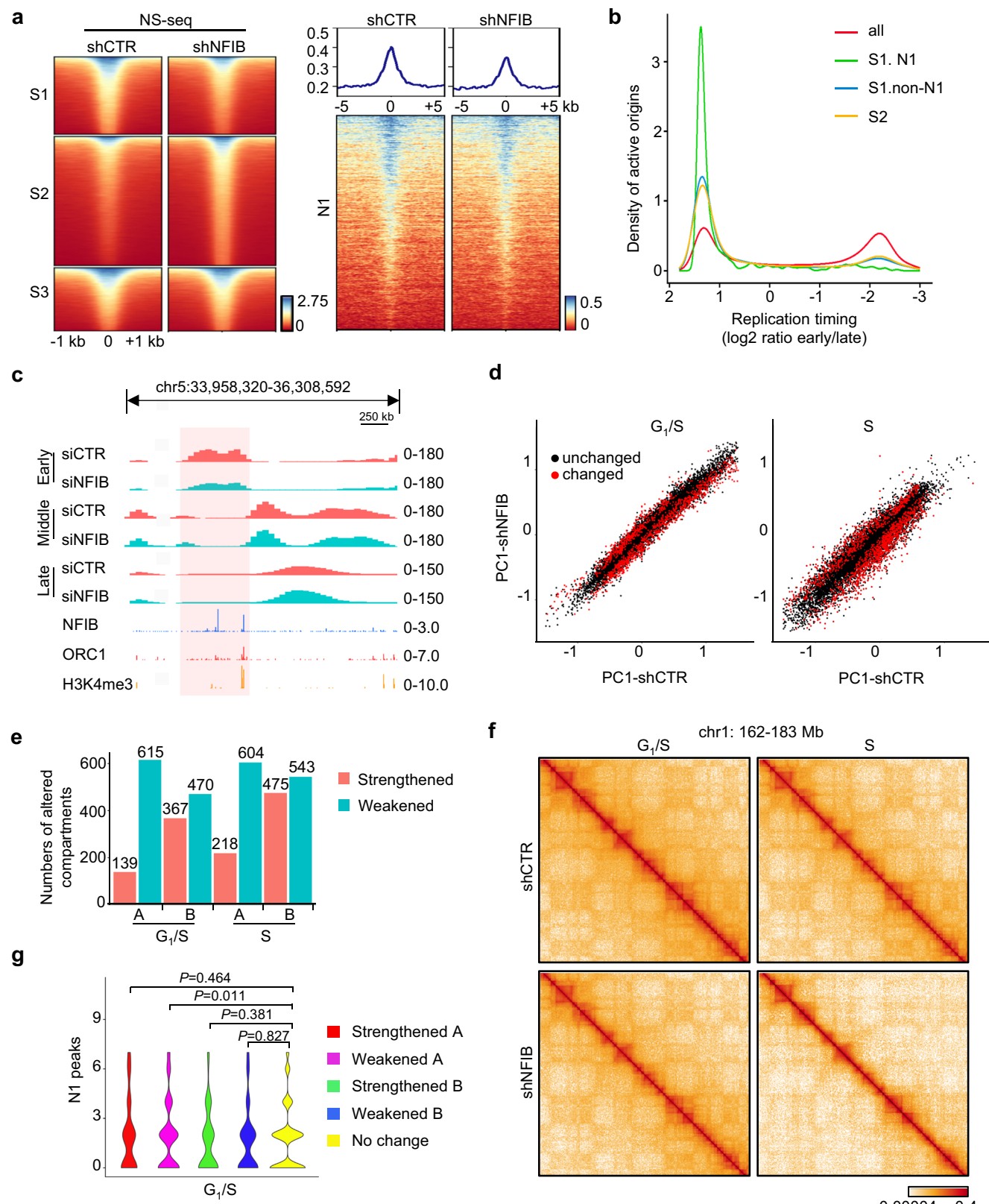

immunoprecipitation (qChIP) assays with antibodies against NFIB, ORC1, CDC6, CDT1, or MCM7 showed that all these proteins were enriched at the two origins (Fig. 4b). Moreover, qChIP analysis of the temporal binding profile showed that the chromatin binding of NFIB preceded the assembly of the pre-RC on the origins during the cell cycle progression from $G_2/M$ to $G_1$ phase (Fig. 4c). To reinforce this

notion, NFIB or the components of the pre-RC was individually depleted in U2OS cells by lentivirally-delivered shRNA or siRNA. Notably, while NFIB knockdown resulted in an overt reduction of the enrichment of ORC1, CDC6, CDT1, and MCM7 on *ORI1*, *ORI2*, and a dormant origin lacking nascent strand signal (Fig. 4d and Supplementary Fig. 3a), it had no effects on the pre-RC assembly in an origin

**Fig. 3 | Depletion of NFIB alters chromosome contacts/compartments and replication profile. a** Left: Heatmap of NS-seq signals in NFIB-depleted versus control U2OS cells. The fold change cutoff of signal intensity was set to 1.2. S1, peak signals decreased; S2, peak signals increased; S3, peak signals unaltered. Right: Heatmap of NS-seq signal distribution surrounding N1 peaks in NFIB-depleted versus control U2OS cells. **b** Different time points of synchronized NFIB-depleted U2OS cells were analyzed by Repli-seq. Control or NFIB-depleted U2OS cells were synchronized via thymidine/aphidicolin block followed by release for 0.5 h, 4 h, or 6 h to enrich cells in the early, middle, or late S phase, respectively. Graph showing the replication timing of NFIB-associated active origins (S1 peaks overlapped with N1 peaks, S1.N1, $n = 415$), other S1 peaks (S1.non-N1, $n = 14,203$), S2 peaks

$(n = 24,410)$ and total NS-seq peaks (all, $n = 48,273$). **c** Comparison between replication initiation patterns after NFIB was depleted in U2OS cells. Genome tracks showing the signals of Repli-seq, NFIB ORC1, and H3K4me3 at selected regions of chromosome 5. **d** Scatter plot of PC1 eigenvector of $G_1$/S boundary or S phase U2OS cells without (shCTR) or with NFIB depletion (shNFIB) for each 250 kb genome bin. Red points: changed regions. Black points: unchanged regions. **e** Histogram showing the number of regions (250 kb bin) where the PC1 eigenvector significantly changed. **f** Knight-Ruiz (KR) balanced Micro-C contact map of chromosome 1: 162 to 183 Mb at a 100-kb resolution. **g** Violin plot showing the distribution of N1 peaks in different chromatin regions in $G_1$/S phase (bin size 250 kb).

lack of NFIB binding or a non-origin with only ORC1 binding (Supplementary Fig. 3b and 3c). On the other hand, while depletion of ORC1, CDC6, CDT1, or MCM7 resulted in a diminished chromatin binding of the corresponding proteins on *ORI1* and *ORI2*, deficiency of these factors had little effect on NFIB binding (Fig. 4d). The knockdown efficiency of individual proteins was verified by western blotting (Fig. 4e). Together, these observations support that NFIB is required for the pre-RC assembly and the subsequent replication initiation at a subset of origins.

### NFIB facilitates nucleosome eviction on parental chromatin

To gain further understanding of NFIB-promoted assembly of the pre-RC and replication initiation, we next took advantage of the transmission electron microscopy (EM)-based approach[52] to directly visualize NFIB-associated chromatin changes during replication. To this end, NFIB-depleted U2OS cells were synchronized at $G_1$/S boundary by double-thymidine block and released 4 h to allow the cell entering S phase. Transmission EM showed that NFIB depletion was associated with a marked increase in the nucleosome density in parental strands, whereas in daughter strands, the nucleosome density slightly decreased (Fig. 5a, b), suggesting that NFIB facilitates parental histone eviction.

To support this notion, we then tested whether or not NFIB could directly interact with nucleosomes using in vitro reconstitution system. To this end, recombinant NFIB was purified from eukaryotic Sf9 cells using the baculovirus expression system (Fig. 5c), and mono-nucleosomes were reconstituted onto a 213-bp DNA fragment containing a single Widom 601 nucleosome-positioning sequence and a NFIB-binding motif. We constructed three types of mono-nucleosomes, B1, B2, and B3, with the NFIB motif positioned on H2A-H2B, H3-H4, and H1, respectively (Fig. 5d). These nucleosomes were then incubated with an increased amount of recombinant NFIB, and the binding preference was examined by running the products on 5% native PAGE gels. The results showed a similar binding efficiency of NFIB to all three types of nucleosomes (Fig. 5e, left). Moreover, the deletion of the NFIB binding motif from the DNA fragment did not affect the binding of NFIB to the mono-nucleosomes (Fig. 5e, right). We further performed histone binding assays with bacterially purified GST-NFIB and native calf thymus histones (CTH) or recombinant *Xenopus* octamers, which represent histones with or without post-translational modifications, respectively. The results from these experiments indicated that NFIB was able to bind H2B and H3, irrespective of histone modification status (Fig. 5f). These observations clearly point to the direct binding of NFIB to nucleosomes, favoring the argument that NFIB facilitates parental histone eviction.

To gain further support for this notion, we next conducted single-molecule magnetic tweezers to trace the structural transition of mono-nucleosomes with or without NFIB depletion to examine how NFIB affects nucleosome dynamics. In these experiments, mono-nucleosomes were reconstituted onto a 409-bp DNA fragment containing a single Widom 601 sequence. Two ends of the DNA fragment were modified with digoxin and biotin for specifically tethering to coverslips and paramagnetic beads, respectively. The

real-time trajectory of individual nucleosomes was traced with a continuously increased tension up to 30 pN. In the presence of NFIB, the nucleosome was completely disassembled at a force of around 5 pN, and the unfolding process was maintained during repeated stretching measurements with magnetic tweezers, whereas in the absence of NFIB, the nucleosome displayed a well-characterized irreversible two-step unfolding dynamics, consistent with the previous reports[53–56]. Two typical extension jumps in the first cycle were observed at a force around 3 pN and 23 pN, which disrupts the outer and inner DNA wrap of the nucleosome, respectively, and these jumps were not observed in repeated stretching measurements, indicating that histones are displaced from DNA after nucleosome was fully disrupted. These observations suggest that NFIB functions not only to destabilize nucleosome structure but also to hold histone octamer onto DNA to maintain the reversibility in nucleosome formation (Fig. 5g). The function of NFIB in the organization of chromatin structure was further supported by qChIP assays showing that knockdown of NFIB led to significant retention of H3 at *ORI1* and *ORI2*, concomitant with a decreased enrichment of ORC1 (Fig. 5h), whereas overexpression of NFIB in normal mammary epithelial MCF-10A cells was associated with an overt decrease in the enrichment of H3 as well as an increased enrichment of ORC1 at *ORI1* and *ORI2* (Fig. 5h). Collectively, these results indicate that NFIB functions to facilitate parental histone eviction, leading to increased chromatin accessibility, thereby promoting the assembly of the pre-RC at target origins.

### NFIB promotes S-phase progression and cell proliferation

To further support the role of NFIB in DNA replication and the associated cellular function of NFIB, we next tested the effect of NFIB on cell cycle and cell proliferation. To this end, U2OS cells with or without knockdown of NFIB were treated with hydroxyurea (HU), followed by BrdU labeling to examine single-stranded DNA (ssDNA) patches formed in S phase (PCNA positive cells) without DNA denaturation[46]. The results showed that depletion of NFIB significantly compromised HU-induced ssDNA formation (Fig. 6a). In addition, U2OS cells with or without NFIB depletion were synchronized at $G_1$/S boundary by double-thymidine block before release for different hours. EdU incorporation for DNA synthesis and cell cycle analysis showed that NFIB-depletion led to either defective S phase progression or arrestment at $G_1$ phase (Fig. 6b, c). These observations are consistent with a function of NFIB in DNA replication.

We then conducted cell counting kit 8 (CCK-8) as well as colony formation assays in U2OS cells to test the effect of NFIB on cell proliferation. These experiments showed that knockdown of NFIB led to inhibited cell proliferation (Fig. 6d) and reduced colony formation (Fig. 6e). U2OS cells were further treated with a low dose of HU to induce replication stress, as it has been reported that insufficient origin licensing could render cells more sensitive to stress conditions[57,58]. We found that HU treatment exacerbated NFIB depletion-associated inhibition of cell proliferation (Fig. 6d) and reduction of colony formation (Fig. 6e), consistent with a role of NFIB in the pre-RC assembly

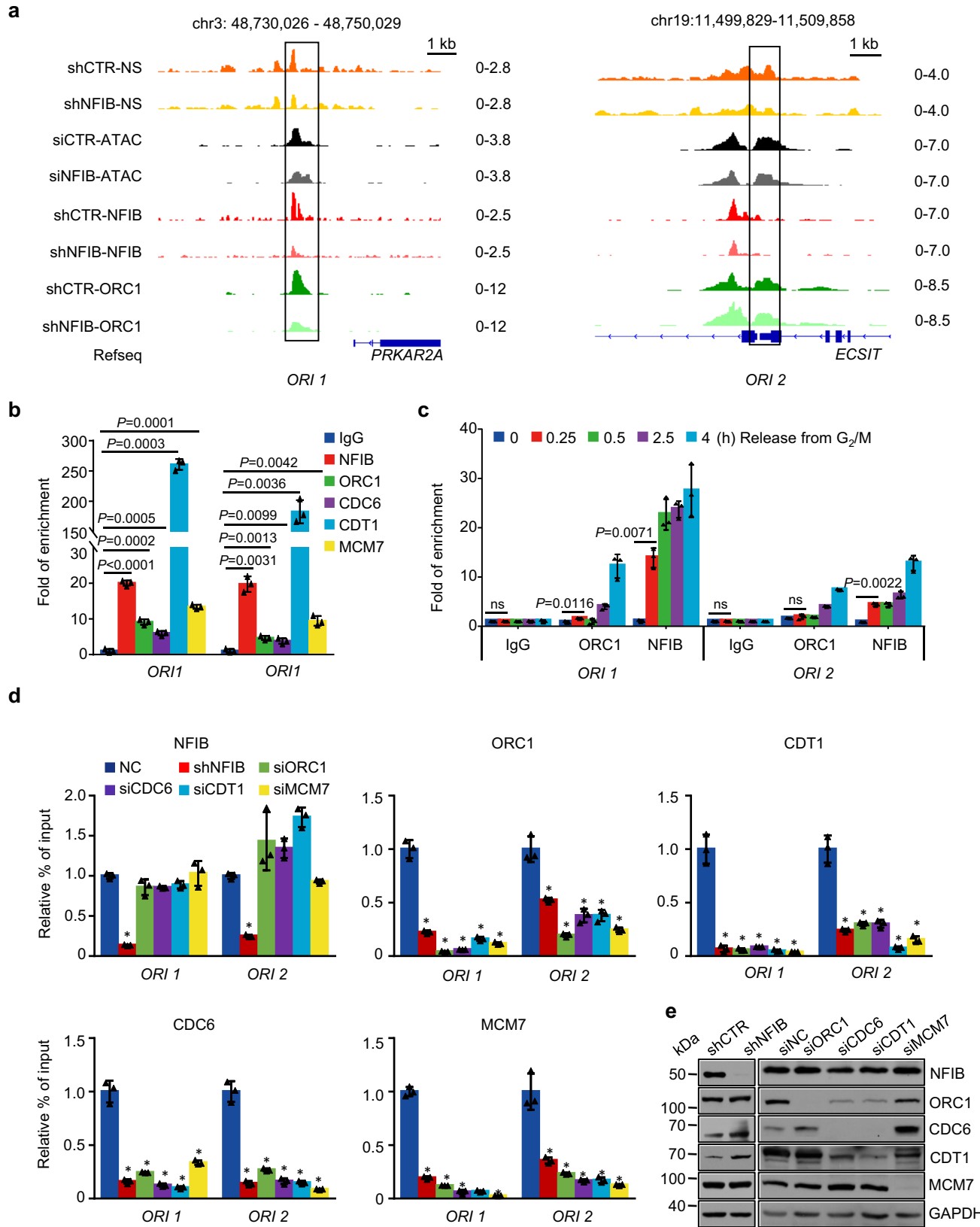

thus origin licensing. On the other hand, overexpression of NFIB in MCF-10A, a non-tumorigenic epithelial cell line with a low level of endogenous NFIB, was associated with accelerated cell proliferation (Fig. 6f) and increased colony formation (Fig. 6g). Together, these observations reinforce the role of NFIB in the pre-RC assembly in replication.

## NFIB overexpression provokes genomic aberrations recapitulating genomic aberrations in breast cancer

Ectopic- or over-licensing of replication origins is linked to replication stress, re-replication, and genomic instability in cancer[59,60]. As stated earlier, *NFIB* is frequently amplified/overexpressed in many types of cancer and is defined as a "cancer-related gene"[28]. To further support

**Fig. 4 | The binding of NFIB and the Pre-RC on chromatin. a** Genome tracks show the signals of nascent strands, chromatin accessibility, NFIB, and ORC1. *PRKAR2A* and *ECSIT* are the Refseq gene names. **b** qChIP verification of chromatin binding of NFIB and pre-RC at the origins shown in A using antibodies against the indicated proteins. The data are presented as the means ± SD for triplicate experiments. *P* values were determined by an unpaired two-tailed *T* test. **c** U2OS cells were synchronized at the G$_2$/M phase and released for different hours. qChIP experiments were performed with antibodies against the indicated proteins to examine the chromatin binding dynamics of NFIB and ORC1. Error bars represent mean ± SD for triplicate experiments. *P* values were determined by an unpaired two-tailed *T* test. **d** U2OS cells were infected with lentiviruses carrying the indicated shRNAs or transfected with the indicated siRNAs. qChIP experiments were performed using antibodies against the indicated proteins. Error bars represent mean ± SD for triplicate experiments (\**P* < 0.0001, 2-way ANOVA). **e** Knockdown **e**fficiency of the indicated factors was verified by western blotting.

NFIB-promoted replication and to investigate whether the oncogenic potential of NFIB is linked to its function in replication, we first performed FACS analysis to examine the cell cycle profile upon NFIB overexpression in the normal mammary epithelial MCF-10A cells. The results showed that the percentage of S-phase, as well as re-replication populations of cells, increased in NFIB-overexpressing cells (Fig. 6h). We additionally performed cesium chloride gradient centrifugation assay to analyze DNA re-replication in MCF-10A cells with or without NFIB overexpression[61]. The results revealed that NFIB-overexpressing cells have higher level of re-replicated (heavy/heavy) DNAs compared to that in control cells (Fig. 6i). While the extent of NFIB-elicited re-replication was limited, these observations were consistent with a role for NFIB in promoting DNA replication.

To gain a general picture of NFIB-affected genome stability, we then established MCF-10A clones stably overexpressing NFIB via a lentivirally delivered NFIB expression vector. These clones were further cultured 30 or 75 days before being subjected to whole genome sequencing. After subtracting spontaneous copy gains in control cells, NFIB-overexpressing MCF-10A clones exhibited more genome amplifications reflected by an increased number of loci with copy gain in clones of 75 days and an increased number of loci with structure insertion (fragments larger than 1 kb[62]) in clones of both 30 days and 75 days (Fig. 7a).

To extend our observations to clinicopathologically relevant settings, we next analyzed the genomic aberrations observed in NFIB-overexpressing cells in reference to the genetic abnormalities in breast cancer samples. We compared recurrent copy number aberrations (CNAs) in NFIB-overexpressing clones of 75 days to CNAs in control clones of 75 days and extracted a total of 1012 NFIB-associated amplification regions. We then retrieved genes co-amplified with NFIB from 4 breast cancer cohorts, including 2051 patient samples in METABRIC (Molecular Taxonomy of Breast Cancer International Consortium) and ~1000 patient samples in TCGA (The Cancer Genome Atlas) through cBioPortal[63–65]. Significantly, ~46% NFIB-associated CNA loci detected in our system were also found in at least one of the four breast cancer cohorts, with a co-occurrence FDR less than 0.05 (Fig. 7b). Functional classification revealed that the top 20 co-amplified genes were known to be critically involved in the development and progression of cancer, including those that are implicated in tumor growth, epithelial-mesenchymal transition (EMT), or angiogenesis (Fig. 7c). For example, the most frequently co-amplified gene *ADORA1* encodes for a protein shown to promote chemotaxis and angiogenesis in melanoma and increase proliferation in breast cancer[66]; TSTA3 has been reported to drive breast cancer cell proliferation and invasion[67]; ARID4B has been documented to promote breast cancer growth and metastasis[68] as a potential prognostic marker[69]; amplification of *ERBB2* occurs in various types of cancer including 15–20% of breast cancer[70] and *ERBB2* is a well-established oncogene[71–74]; FOXA1 is the first identified transcription pioneers and constitutes a major proliferative and survival axis for estrogen receptor positive (ER$^+$) breast cancer[75–77]. Amplification of selected genes deduced from the above analysis was experimentally confirmed by Taqman copy-number assay performed with MCF-10A cells with or without NFIB overexpression (Fig. 7d). Collectively, these observations indicate that at least in breast cancer cells, NFIB overexpression has a potential to cause genomic aberrations that could ultimately lead to carcinogenesis, supporting the role of NFIB in genome organization and replication initiation.

## NFIB-provoked genome aberrations evolve to confer therapeutic resistance in breast cancer cells

The occurrence of genomic alterations associated with NFIB overexpression and the clinicopathological relevance of these genetic abnormalities suggest an evolving and inheritable theme along with NFIB-promoted carcinogenesis. To further extend our observations to clinicopathologically relevant settings, CCK-8 assays in the MCF-10A clones of 75 days revealed that NFIB-overexpression was associated with chemoresistance of the cells to multiple chemotherapeutic compounds including epirubicin, nocodazole, oxaliplatin, and 5-fluoracil (Fig. 7e). To investigate whether NFIB-associated chemoresistance is mechanistically linked to NFIB-evoked genomic aberrance, we further performed whole genome sequencing in NFIB-overexpressing/drug-resistant MCF-10A cells under the treatment of oxaliplatin and 5-fluoracil. Compared to control cells, NFIB-overexpressing MCF-10A cells treated with 5-fluoracil and oxaliplatin generated additional sets of CNAs, especially amplified CNAs (Fig. 7f and Supplementary Fig. 4). Comparative analysis revealed a total of 1290 additionally amplified CNAs in NFIB-overexpressing/5-fluoracil-resistant cells and 1035 additionally amplified CNAs in NFIB-overexpressing/oxaliplatin-resistant cells (Fig. 7g). Functional annotation of the genes associated with additionally amplified CNAs in NFIB-overexpressing/drug-resistant cells yielded indeed sets of genes that are known to play key roles in chemoresistance, including AKT1, ALDH3A1, ASNA1, and multiple members of the ABC transporter family (Supplementary Fig. 4). These observations support a notion that NFIB-evoked genomic aberrance empowers cancerous cells to evolve fitness of growth advantages under selective pressures by chemotherapeutics.

As stated above, *ERBB2* is one of the co-amplified genes in NFIB-overexpressing MCF-10A cells (Fig. 7c and Supplementary Fig. 4). To confirm the co-amplification of *ERBB2* and NFIB overexpression and to support the clinicopathological significance of this co-amplification in breast carcinogenesis, we collected a panel of breast cancer cell lines for western blotting analysis of the expression of NFIB. Among them, SK-BR-3 and BT474 express high levels of HER2 with *ERBB2* gene amplification thus labeled as HER2$^+$, ZR-75-1 cells express HER2 without *ERBB2* gene amplification and thus labeled as HER2$^{+/-}$, and other cells were considered HER2$^-$[78]. High levels of NFIB expression were found in the two HER2$^+$ cells as well as in ZR-75-1 cells, which are HER2$^{+/-}$, and MCF-7 cells, which are HER2$^-$ (Fig. 8a). Since co-amplification of *ERBB2* is not the only NFIB-evoked genomic aberration, it is possible that NFIB-elicited gene amplification leans to other loci in ZR-75-1 and MCF-7 cells. We then repeated a series of biochemical and cellular experiments in two NFIB$^{high}$ breast cancer cell lines, ZR-75-1 and BT-474, to validate the interactions between NFIB and pre-RC (Fig. 8b), NFIB-dependent chromatin loading of pre-RC (Supplementary Fig. 5a), and NFIB-associated cell cycle arrestment (Fig. 8c). Furthermore, CCK-8 assays in BT-474 or ZR-75-1 cells revealed that NFIB-deficiency was associated with increased drug sensitivity to epirubicin, nocodazole, oxaliplatin, and 5-fluoracil (Fig. 8d and Supplementary Fig. 5b).

We then performed Kaplan–Meier survival analysis with different subtypes of breast cancer patients (log-rank test). The results showed

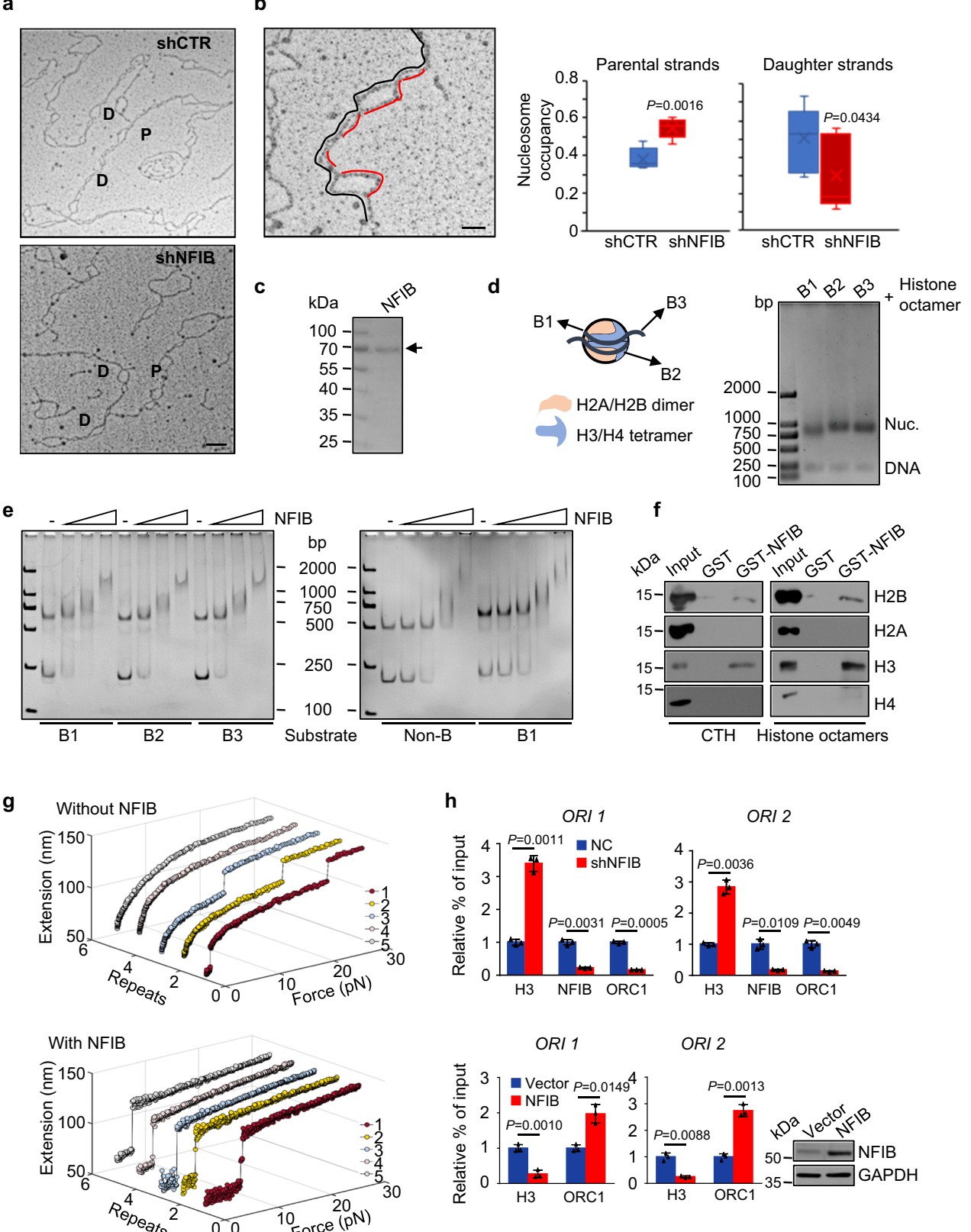

that higher NFIB expression was correlated with a worse distant metastasis-free survival (DMFS) as well as with relapse-free survival (RFS) of breast cancer patients of HER2[+] subtypes, but not of other subtypes (Fig. 8e and Supplementary Fig. 6a). Multivariate COX regression analysis incorporated with cell proliferation marker Ki-67 also showed that high NFIB expression is correlated with worse DMFS

(HR = 1.46, 95% CI: 1.05–2.05; $p$ = 0.03) and worse RFS (HR = 1.46, 95% CI: 1.17–1.82; $p$ < 1e-04) of HER2[+] breast cancer patients but not with patients of other subtypes (Supplementary Fig. 6b), a result consistent with the univariate survival analysis (log-rank test). While the correlation analysis supports the role of NFIB in HER2[+] breast carcinogenesis, it does not exclude its function in other subtypes of breast cancer or

**Fig. 5 | NFIB facilitates nucleosome eviction and nucleosome disassembly.**
**a** Analysis of psoralen cross-linked replication intermediates by EM. Representative images are shown. P and D denote parental and daughter strands, respectively. Scale bar, 200 nm. **b** Nucleosome occupancy was calculated from the combined contour length of all nucleosome bubbles in a given stretch of DNA (red), divided by the overall contour length of the DNA duplex (black). Scale bar, 50 nm (≈125 bp) (left). Nucleosome occupancy on parental (N = 5) and daughter DNA strands (N = 8). The median is displayed. Boxes are 25–75 percentile ranges, and whiskers are 0–100 percentile ranges (right). The data are presented as the means ± SD. *P* values were determined by an unpaired two-tailed *T* test. The experiment was performed three times with similar observations. **c** SDS-PAGE of purified NFIB protein using the baculovirus expression system. The experiment was performed three times with similar observations. **d** Schematic diagrams of the three types of NFIB motif-containing nucleosomes (left) and their effective reconstitution showing by the gel electrophoresis mobility shift assay (right). The experiment was performed three times with similar observations. **e** Nucleosome binding assays were performed with

0.2, 0.4, or 1 μg NFIB (left) or with 0.1, 0.2, 0.4, or 1 μg NFIB (right) and 0.1 μg of the indicated nucleosomes. The reaction was analyzed by gel electrophoresis. The experiment was performed three times with similar observations. **f** In vitro histone binding assays with bacterially expressed GST-fused proteins and calf thymus histones (CTH) or recombinant *Xenopus* histone octamers. Bound histones were detected by western blotting using anti-H2B and anti-H3 antibodies. The experiment was performed three times with similar observations. **g** Repeated stretching measurements of mono-nucleosome without or with NFIB. In each stretching cycle, the force was elevated up to 32 pN at a loading rate of 0.1 pN/s. **h** U2OS cells (upper) were infected with lentiviruses carrying the indicated shRNA, and MCF-10A cells (lower) were transfected with control or NFIB expression plasmids for qChIP assays on NFIB-associated origins using antibodies against the indicated proteins. Overexpression efficiency was verified by western blotting. The data are presented as the means ± SD for triplicate experiments. *P* values were determined by an unpaired two-tailed *T* test.

other cancers considering the complex molecular signature and tumor heterogeneity in those cases.

## Discussion

Precise DNA replication is of fundamental importance for genome stability and systemic homeostasis. Although the selection and activation of replication origins in metazoans are still under intensive investigations, evidence is accumulating to suggest that certain chromatin signatures, rather than consensus DNA motifs, instruct replication origin selection/activation by influencing chromatin architecture and shaping the genomic landscape of replication[15,17,18,79]. In this study, we report that NFIB, a member of NFI family that has been mainly investigated as a transcription factor, is physically associated with the DNA replication machinery and functionally promotes the assembly of the pre-RC on chromatin. We demonstrated that NFIB directly binds to nucleosomes and facilitates parental histone eviction, leading to increased chromatin accessibility as a genome organizer to generate chromatin environments conducive to origin licensing. In this sense, NFIB is a replication pioneer more or less analogous to transcription pioneer factors such as FOXA1[80]. The analogy between NFIB in replication and a pioneer factor in transcription is drawn based on the following features: first, pioneer factors target specific genome sites, by their name, prior to the loading of transcriptosome or replicosome; second, pioneer factors are able to bind directly to nucleosomes in vivo and to reconstituted mononucleosomes in vitro; third, pioneer factors are capable of increasing chromatin accessibility and often functionally associated with chromatin openness.

However, while transcription pioneer factors often affect a large scale of chromatin and are functionally linked to cell programming and reprogramming, the extent of NFIB in affecting origin firing and genomic duplication is limited, reflecting the more complex characteristics of DNA replication, in which all genetic materials must be duplicated during each cell cycle and ceasing one origin is often compensated by firing an alternative one[1–3]. Replication occurs in all cell lineages and involves profoundly a larger scale of the genome than gene transcription, which is rather more selective and lineage specific. Only a subset of replication origins are used to replicate the eukaryotic genome at each cell cycle[22]; dormant origins are rarely used under normal conditions but can be activated in specific cellular programs or under certain cellular conditions[23]. Thus, DNA replication intuitively requires more sophisticated regulation and coordination. Considering the existence of a plethora of transcription pioneers, the existence and implementation of even more replication pioneers in eukaryotic cells are not unexpected.

The molecular details involved in the genome organization by NFIB still need investigation. However, the notion that NFIB facilitates histone eviction is consistent with our observations that NFIB is also co-purified with FACT, a prominent histone chaperone complex that is

well characterized for its role in nucleosome remodeling via eviction or assembly of histones[55,81,82]. The notion of NFIB as a genome organizer is also consistent with the recent report that NFIB is functionally linked to a widespread increase in chromatin accessibility, although the extent of the influence can vary in different cell types and under different conditions[36].

We showed that cancer-associated NFIB overexpression provokes genetic alterations that mimic the genomic aberrations in breast carcinomas and that NFIB-evoked genomic abnormalities dynamically evolve to confer growth advantage and therapeutic resistance. While overexpression of NFIB in normal mammary epithelial MCF-10A cells caused limited re-replication as shown by the FACS and cesium chloride gradient centrifugation assay, many a mickle makes a muckle, such small changes would eventually lead to cancerous genomic aberrations including amplification of oncogenes such as *ERBB2*, as shown by the genome sequencing analysis in the long-term cell cultures. As our study is by no means to exclude the function of NFIB in gene transcription, the replication pioneering activity of NFIB could be a functional result of transcription-replication coupling[83], reflected by the fact that NFIB-affected origins are mainly enriched in early-replication domains in open chromatin. In this regard, it is worth noting that c-Myc, a prototypical oncogene best known as a transcription factor, also has been reported to control DNA replication by interacting with the pre-RC complex[84]. Whether c-Myc can be considered as a replication pioneer factor needs further investigation, it is nonetheless evident that transcription-replication coupling is one of the major themes of chromatin-associated activities in eukaryotes[85]. It is even intriguing to speculate that the replication regulatory function of the overreplication/genomic aberrance associated with the involving factors, the potential replication pioneers, might also be the oncogenic driving force behind some of the well-established transcription factors. Clearly, more investigations are warranted for replication priming/pioneering and transcription-replication coupling.

Interestingly, *NFIB* is the only member of the NFI family that is classified as "cancer-related genes" in the Human Protein Atlas. Indeed, overexpression/amplification of NFIB is frequently found in cancers from multiple tissue origins, including lung, skin, breast, and bone[28], whereas to the best of our knowledge, the pathophysiological functions of NFIA, NFIC, and NFIX remain poorly understood and their oncogenic potentials remain to be investigated. In addition, it has been reported that overexpression of NFIB triggers human colon carcinoma HCT116 cells to enter the S phase[86], consistent with our current model that NFIB functions in replication. Moreover, the oncogenic potential of NFIB has been best studied in SCLC, where a number of groups described its role in cancer initiation and metastasis using genetically engineered mouse models and human patient specimens[28,35,36]. It is logical to propose that NFIB-associated erroneous DNA replication and genomic instability might, at least partially, underlie the oncogenic

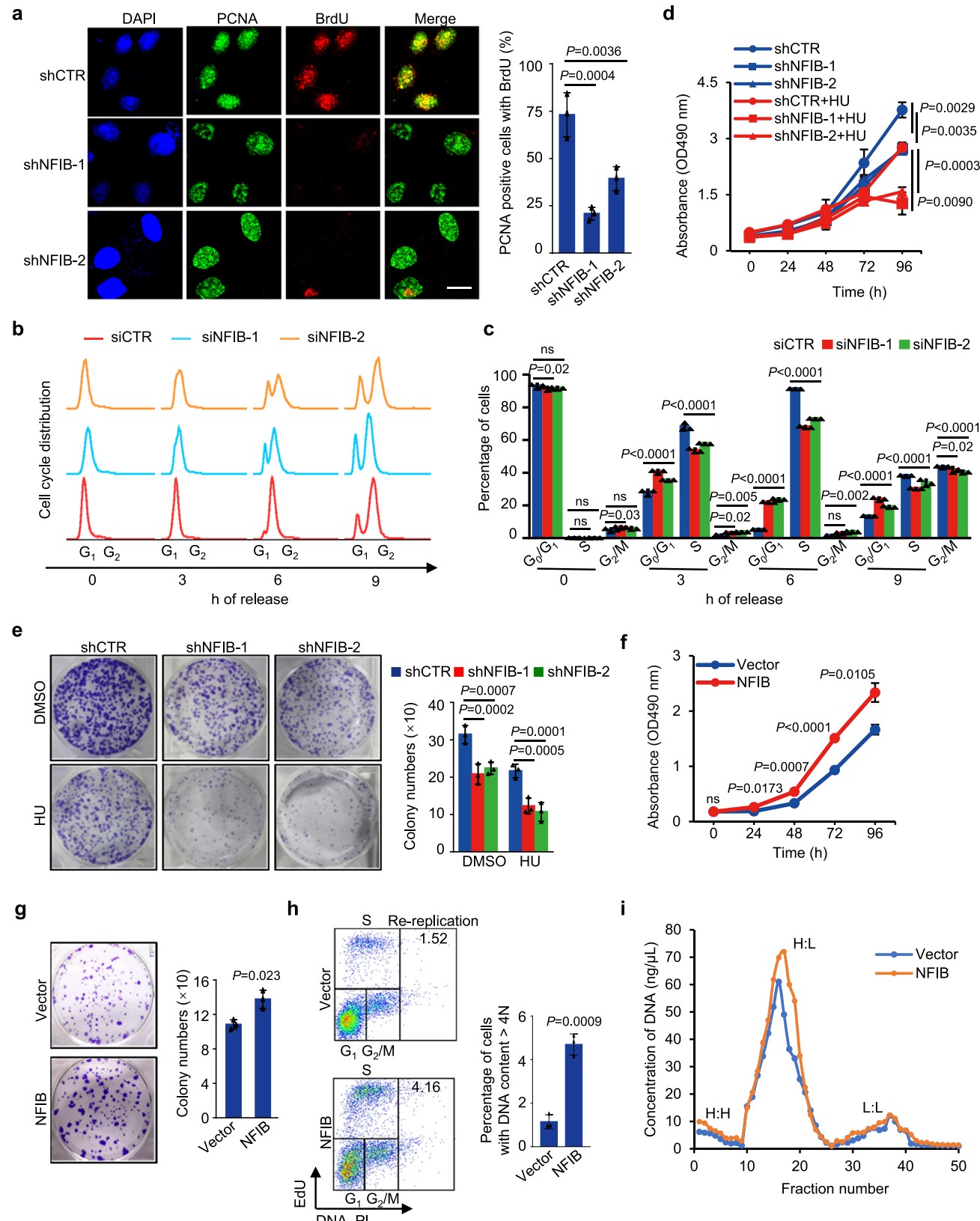

potential of NFIB in various types of cancers, including SCLC. However, intriguingly, the same study in HCT116 cells reported that overexpression of other NFI members including NFIA, NFIC, and NFIX rather led to a reduced percentage of S-phase cells[86]. As mentioned earlier, it is believed that the NFI proteins act as homo- or heterodimers[24], and we indeed detected NFIC and NFIX in the NFIB interactome. Although it is not out of the scope of our current investigation, due to the large volume of this study, we did not investigate the functional relationship between NFIB and NFIC/NFIX and the possible involvement of NFIC and NFIX in genome organization and replication pioneering of NFIB. Nevertheless, our study proposes NFIB being a genome organizer through facilitating nucleosome

**Fig. 6 | NFIB promotes S-phase progression and cell proliferation. a** ssDNA formation in control or NFIB-depleted U2OS cells after HU treatment. BrdU (20 μg/ml) was added during the last 24 h before cells were harvested and removed by a brief wash prior to HU treatment (4 mM, 2 h), pre-extraction, and fixation. BrdU in ssDNA patches and PCNA were detected without DNA denaturation. Scale bar, 10 μm (left). Diagrams show the frequency of PCNA-positive BrdU foci (right). The data are presented as the means ± SD for triplicate experiments. *P* values were determined by one-way ANOVA. **b**, **c** Synchronized U2OS cells were analyzed at different time points after release from G$_1$/S transition for EdU incorporation by flow cytometry. The data are presented as the means ± SD for triplicate experiments. *P* values were determined by two-way ANOVA followed by the Dunnett test. NS, not significant. **d** U2OS cells were treated with or without 200 μM HU for CCK-8 assays. The data are presented as the means ± SD for three independent experiments. *P* values were determined by an unpaired two-tailed *T* test. **e** U2OS cells were infected with lentiviruses carrying the indicated shRNA and cultured in regular medium or medium supplemented with 200 μM HU for 14 days. Cells were then stained with crystal violet and counted for colony numbers. The data are presented as the means ± SD for three independent experiments. *P* values were determined by

two-way ANOVA followed by the Dunnett test. **f** CCK-8 assays in MCF-10A cells infected with lentiviruses carrying control vector or NFIB expression construct. The data are presented as the means ± SD for three independent experiments. *P* values were determined by an unpaired two-tailed *T* test. **g** MCF-10A cells were infected with lentiviruses carrying control or NFIB expression construct. Colony formation assays were performed. The data are presented as the means ± SD for three independent experiments. *P* values were determined by an unpaired two-tailed *T* test. **h** FACS analysis of EdU incorporation and DNA content after 1-h pulse with EdU. Re-replicated DNA ( >4 N) contents are presented as gates, and their percentages are plotted as bars. The data are presented as the means ± SD for three independent experiments. *P* values were determined by an unpaired two-tailed *T* test. **i** CsCl density gradient analysis for the replication pattern of BrdU-labeled genomic DNAs in NFIB-overexpressing MCF-10A cells. Data were presented as DNA concentration of the indicated fraction from the bottom of the CsCl gradient. The fractions showing light: light (L:L), heavy: light (H:L), or heavy: heavy (H:H) DNA content were indicated. A representative result from three independent repeat experiments is shown.

remodeling and a replication pioneer via promoting the assembly of pre-RC. We also propose that cancer-associated NFIB overexpression/amplification could lead to replication alterations and genomic aberrations, eventually to the development of cancer. The current study might add to the understanding of the pathophysiological function of NFIB in specific and to the understanding of the selection and activation of eukaryotic replication origins in general.

## Methods

### Antibodies and reagents

Antibodies used and the sources are as follows: NFIB (ab186738, 1:1000 for WB), ORC1 (ab85830, 1:1000 for WB), CDC6 (ab188423, 1:1000 for WB), CDT1 (ab70829, 1:1000 for WB), MCM5 (ab76023, 1:1000 for WB), MCM6 (ab201683, 1:1000 for WB), MCM7 (ab52489, 1:1000 for WB), PCNA (ab29, 1:2000 for WB), H3 (ab1791, 1:2000 for WB) and BrdU (anti-CldU, ab6326) from Abcam; MCM2 (3619, 1:1000 for WB) from Cell Signaling Technology; ORC2 (sc-32734, 1:500 for WB), Geminin (sc-74456, 1:1000 for WB), POLD1 (sc-17776, 1:1000 for WB) and β-actin (sc-47778, 1:2000 for WB) from Santa Cruz Biotechnology; MCM4 (13043-1-AP, 1:1000 for WB), Cyclin E1 (11554-1-AP, 1:1000 for WB) and MCM3 (A1060, 1:1000 for WB) from Abclonal; NFIB (A303-566A, for CUT&Tag) and ORC1 (A301-892A, for CUT&Tag) from Bethyl Laboratories; c-myc-Tag (B1022, 1:2000 for WB), HA-Tag (B1021, 1:2000 for WB) and GAPDH (B1034, 1:2000 for WB) from Biodragon. FITC or TRITC-conjugated secondary antibodies (ZF-0311, ZF-0312, ZF-0316, ZF-0313, 1:100 for IF) from ZSGB-BIO.

### Cell culture and transfection

Cell lines used were obtained from the American Type Culture Collection (ATCC). U2OS and HEK293T cells were maintained in DMEM supplemented with 10% FBS in a humidified incubator equilibrated with 5% CO$_2$ at 37 °C. MCF-10A cells were maintained in F-12 supplemented with 10% horse serum, insulin, cholera toxin, EGF, and hydrocortisone in a humidified incubator equilibrated with 5% CO$_2$ at 37 °C. BT-474 cells and ZR-75-1 cells were maintained in RPMI-1640 supplemented with 10% FBS in a humidified incubator equilibrated with 5% CO$_2$ at 37 °C. Sf9 cells were cultured in sf-900 II SFM supplemented with 2% FBS, 1% Penicillin/Streptomycin at 27 °C. Transfections were carried out using poly ethylene imine (PEI) (Polysciences) or Lipofectamine RNAiMAX Reagent (Invitrogen) according to the manufacturer's instructions.

### Lentiviral production and infection

Generation of the pLKO.1-shNFIB and pCDH-NFIB lentiviruses was conducted according to a protocol described by Addgene. Briefly, pLKO.1-shNFIB1/2 was generated by subcloning shRNA into the pLKO.1

vector and full length of NFIB coding sequence was subcloned into pCDH. The lentiviral plasmid vector, pLKO.1 or pCDH, pLKO.1-shNFIB1/2 or pCDH-NFIB, together with psPAX2 and pMD2.G were co-transfected into the packaging cell line HEK293T. Viral supernatants were collected 48 h later, clarified by filtration, and concentrated by ultracentrifugation. The concentrated virus was used to infect $5 \times 10^5$ cells (20–30% confluent) in a 60 mm dish with 5 μg/ml polybrene. Infected cells were selected by 2 μg/ml puromycin (Merck). The target sequences were as follows: shNFIB-1, 5′-TGGTTATCTCACCAACGA ACTC-3′; shNFIB-2, 5′-GTTGCCATTTCCAACACAACTC-3′. For the RNAi experiment, the target sequences were as follows: siORC1, 5′-CUGCA-CUACCAAACCUAUAUU-3′; siCDC6, 5′-CUCCAGUGAUGCCAAACUAUU-3′; siCDT1, 5′-AACGUGGAUGAAGUACCCGACUU-3′; siMCM7, 5′-GGCUA AUGGAGAUGUCAAUU-3′.

### Co-immunorecipitation and Western blotting

Cellular lysates were prepared by incubating the cells in lysis buffer (50 mM Tris-HCl, pH 7.5, 150 mM NaCl, 0.3% NP-40, and 2 mM EDTA) containing protease inhibitor cocktail (Roche), followed by centrifugation at 12,100 g at 4 °C for 15 min. The protein concentration of the lysates was determined using the Pierce BCA Protein Assay Kit (Thermo Scientific). For immunoprecipitation, 500 μg of protein was incubated with 2 μg specific antibodies or normal IgG at 4 °C for 12 h with rotation. Protein A or G beads (Thermo Scientific) were added, and the incubation was continued for an additional 2 h. Beads were washed 3 times using the lysis buffer. The precipitated proteins were eluted from the beads by resuspending the beads in SDS sample buffer and boiling for 10 min. The resultant materials from immunoprecipitation or cell lysates were resolved on SDS-PAGE and analyzed by western blotting.

### Synchronization and flow cytometry

To synchronize U2OS cells at G$_1$/S boundary, 2 mM thymidine was added. After 17 h, cells were washed twice with fresh medium, grown for 12 h, and incubated with 2 mM thymidine for an additional 14 h. Similar to U2OS cells, to synchronize BT-474 cells at G$_1$/S boundary, 2 mM thymidine was added. After 14 h, cells were washed twice with preheated PBS, grown for 10 h with fresh medium, and incubated with 2 mM thymidine for an additional 14 h. Cells were then released into a fresh medium, and aliquots were taken every 2 h for flow cytometry and chromatin fractionation. For flow cytometry, cells were collected, washed, and resuspended with cold PBS, and fixed in chilled ethanol overnight. Cells were then washed and resuspended in PBS with 120 μg/ml propidium iodide (PI) and 10 μg/ml RNase A for 30 min at 37 °C. DNA content was measured by flow cytometry and data were analyzed using FlowJo_V10 software.

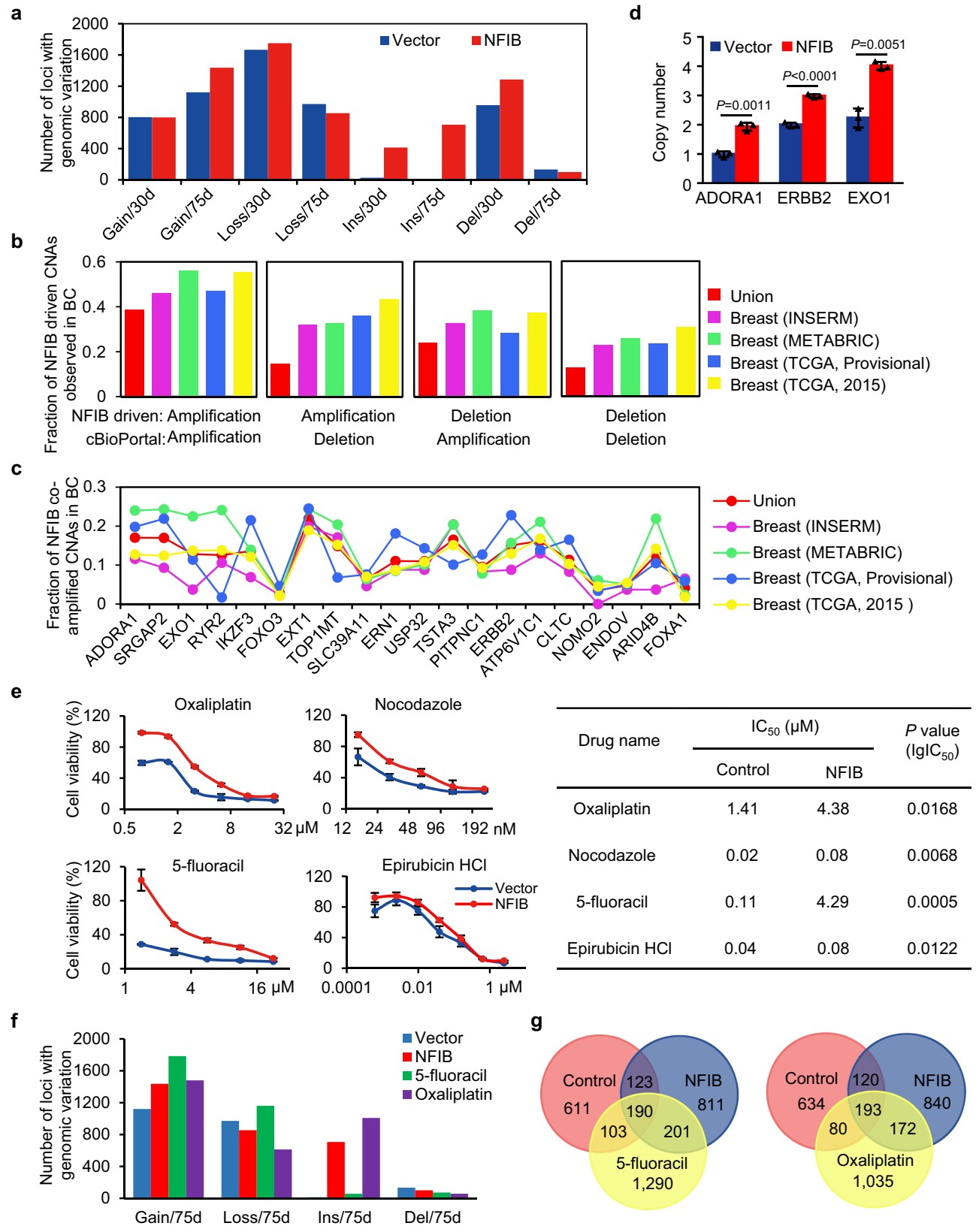

**qPCR**

Total cellular RNAs were isolated from samples with the Trizol reagent (Invitrogen). First strand cDNA synthesis was conducted with the Reverse Transcription System (TransGen Biotech). Quantitation of all gene transcripts was done by qPCR using Power SYBR Green PCR Master Mix and an ABI PRISM 7500 sequence detection system (Applied Biosystems, Foster City, CA). The expression of GAPDH was used as the internal control.

**Cell fractionation and immunoprecipitation**
Chromatin fractionations were extracted as described previously[46]. Briefly, U2OS cells were lysed in CSK buffer (10 mM PIPES pH 7.0,

**Fig. 7 | NFIB overexpression provokes genomic aberrations that mimics genomic aberrations in breast cancer. a** Bar plotting of loci with copy number gain, copy number loss, insertion, or deletion in MCF-10A cells infected with lentiviruses carrying control vector or NFIB expression construct. **b** Fraction of NFIB-associated CNAs observed in breast cancer samples. The union represents events that are observed in any of the tested datasets. **c** Fraction of the alteration in cancer samples of the indicated genes identified from NFIB overexpression-associated amplification that also significantly co-varied with NFIB CNAs in breast cancer samples. **d** Copy number of the indicated genes in MCF-10A cells stably expressing empty vector or NFIB was measured by Taqman copy-number assay. The data are presented as the means ± SD for three independent experiments. *P* values were determined by an unpaired two-tailed *T* test. **e** Control or MCF-10A cells stably expressing NFIB were treated with the indicated anti-neoplastic compounds for 3 days. The growth of cells was measured with the CCK-8 assay. Error bars represent mean ± SD for triplicate experiments. IC$_{50}$ (half maximal inhibitory concentration) was listed. *P* values of lgIC$_{50}$ were calculated based on the extra sum-of-squares *F*-test. **f** Bar plotting of loci with copy number gain, copy number loss, insertion, or deletion in control MCF-10A cells, NFIB-overexpressing MCF-10A cells cultured for 75 days, and NFIB-overexpressing/drug-resistant MCF-10A cells. **g** Venn diagrams of the overlapping and differential sets of amplified CNAs by comparing the data from control MCF-10A cells, NFIB-overexpressing MCF-10A cells cultured for 75 days, and NFIB-overexpressing/drug-resistant MCF-10A cells.

100 mM NaCl, 300 mM sucrose, 3 mM MgCl$_2$) + 0.5% Triton X-100 buffer with rotation at 4 °C for 20 min. After centrifugation for 5 min at 2000 g, the pellet was gently washed twice with CSK buffer and re-suspended in CSK buffer containing DNase I (RQ1 RNase-Free DNase, Promega, M6101) at a final concentration of 250 U/10$^7$ cells. The pellet was digested at room temperature for 15 min at 37 °C for another 15 min, and centrifuged at 12,100 g for 15 min to obtain the supernatant as the chromatin fraction. Immunoprecipitation was then performed as described.

### Silver staining and mass spectrometry
Cellular extracts from U2OS cells expressing FLAG-NFIB were prepared and applied to anti-FLAG M2 affinity gel (Sigma-Aldrich) following the manufacturer's protocol. FLAG peptide (0.2 mg/ml; Sigma-Aldrich) was added to the column to elute the protein complex. Fractions of the bed volume were collected and resolved on SDS-PAGE and silver-stained, and bands were excised and subjected to liquid chromatography–tandem mass spectrometry sequencing and data analysis.

### GST pull-down assay
GST fusion constructs were expressed in BL21 *E. coli* bacteria, and crude bacterial lysates were prepared by sonication in TEDGN (50 mM Tris-HCl, pH 7.4, 1.5 mM EDTA, 1 mM dithiothreitol, 10% (v/v) glycerol, 0.4 M NaCl) in the presence of the protease inhibitor mixture. In vitro, transcription and translation experiments were done with rabbit reticulocyte lysate (TNT systems, Promega) according to the manufacturer's recommendation. Briefly, equal amounts of GST fusion proteins were immobilized on 50 µl of 50% glutathione-Sepharose 4B slurry beads (Amersham Biosciences) in 0.5 ml of GST pull-down binding buffer (10 mM HEPES, pH 7.6, 3 mM MgCl$_2$, 100 mM KCl, 5 mM EDTA, 5% glycerol, 0.5% CA630). After incubation for 1 h at 4 °C with rotation, beads were washed three times with GST pull-down binding buffer and resuspended in 0.5 ml of GST pull-down binding buffer before adding 10 µl of in vitro transcribed/translated proteins for 2 h at 4 °C with rotation. The beads were then washed three times with the binding buffer. The bound proteins were eluted by boiling in 30 µl of 2× sample loading buffer and resolved on SDS-PAGE.

### Immunofluorescent staining
U2OS cells growing on six-well chamber slides were washed with PBS, fixed in 4% (w/v) paraformaldehyde, permeabilized with 0.1% (v/v) Triton X-100 in PBS, or pre-extracted in CSK buffer (10 mM PIPES pH 7.0, 100 mM NaCl, 300 mM sucrose, 3 mM MgCl$_2$) + 0.5% Triton X-100 for 5 min on ice and then fixed with 4% paraformaldehyde. Chilled methanol was applied afterwards if needed, according to the specific requirement for the antibodies. Cells were blocked with 0.8% BSA, and incubated with appropriate primary and secondary antibodies with three times of washes in between. A final concentration of 0.1 µg/ml DAPI (Sigma) was included in the final washing to stain nuclei. Images were visualized with a Zeiss LSM880 fluorescence microscope (Carl Zeiss Inc.).

### Detection of ssDNA
Assays for the detection of ssDNA with immunostaining were performed as previously described[46]. In brief, cells were grown on six-well chamber slides, and BrdU (20 µM) was added during the last 24 h of siRNA (specific for NFIB) treatment and removed by a brief wash prior to HU treatment (4 mM, 2 h). Cells were extracted for 5 min with 0.5% CSK buffer and rinsed with CSK and PBS before being fixed for 10 min in 100% methanol at room temperature. Chamber slides were subsequently washed three times with PBS, treated with cold 70% ethanol at −20 °C for 1 h, blocked with 1% BSA, and incubated with PCNA and BrdU antibodies followed by staining of FITC or TRITC-conjugated secondary antibodies. BrdU in ssDNA patches and PCNA were detected without a DNA denaturation step. Images were visualized and recorded with a Zeiss LSM880 fluorescence microscope (Carl Zeiss Inc.).

### qChIP
U2OS cells were crosslinked using 1% formaldehyde for 10 min at room temperature and quenched by the addition of glycine to a final concentration of 125 mM for 5 min. The fixed cells were resuspended in lysis buffer (1% SDS, 5 mM EDTA and 50 mM Tris-HCl, pH 8.1) in the presence of protease inhibitors, then subjected to 30 cycles (30-s on and off) of sonication (Bioruptor, Diagenode) to generate chromatin fragments of ~300 bp in length. Lysates were diluted in buffer containing 1% Triton X-100, 2 mM EDTA, 150 mM NaCl, 20 mM Tris-HCl (pH 8.1), and protease inhibitors. For IP, the diluted chromatin was incubated with normal IgG (control) or specific antibodies for 12 h at 4 °C with constant rotation, 50 µL of 50% (vol/vol) protein A/G Sepharose beads were then added, and the incubation was continued for an additional 2 h. Beads were washed with the following buffers: TSE I (0.1% SDS, 1% TritonX-100, 2 mM EDTA, 150 mM NaCl and 20 mM Tris-HCl, pH 8.0); TSE II (0.1% SDS, 1% Triton X-100, 2 mM EDTA, 500 mM NaCl and 20 mM Tris-HCl, pH 8.0); TSE III (0.25 M LiCl, 1% Nonidet P-40, 1% sodium deoxycholate, 1 mM EDTA and 10 mM Tris-HCl, pH 8.0) and TE (1 mM EDTA and 10 mM Tris-HCl, pH 8.0). The pulled-down chromatin complex and input were de-crosslinked at 55 °C for 12 h in elution buffer (1% SDS and 0.1 M NaHCO$_3$). The eluted DNA was purified with the QIAquick PCR Purification Kit. qChIPs were performed using Power SYBR Green PCR Master Mix and ABI PRISM 7500 sequence detection system (Applied Biosystems, Foster City, CA). Primers used were: ORI1: 5′-CTGGGCTGAGACTTGGAAGG-3′ (F) and 5′-ACAGCTGCAGTACACACTCC-3′ (R); ORI2: 5′-GAACTAAGCG GGGGAGGATG-3′ (F) and 5′-GTTCCGACTGCCCAATCCTA-3′ (R). Control site in Supplementary Fig. 3a: 5′-TTGTGCTTTCGCTGCAGTTC-3′ (F) and 5′-CTGGCTGCTTCTTTCAGTGG-3′ (R); control site in Supplementary Fig. 3b: 5′-GGACGGGTATGTAATCGGGC-3′ (F) and 5′-GGG GAGACGTTATGGGGTTC-3′ (R); control site in Supplementary Fig. 3c: 5′-CCTTGCTTTGCCACAGGTTC-3′ (F) and 5′-AGGAGCAACAGGGCTC TCTA -3′ (R).

### Histone binding assays
Histone binding assays were performed essentially as described previously[87]. Briefly, recombinant full-length or deletions of NFIB-GST fusion proteins were expressed in *E. coli* strain BL21 and

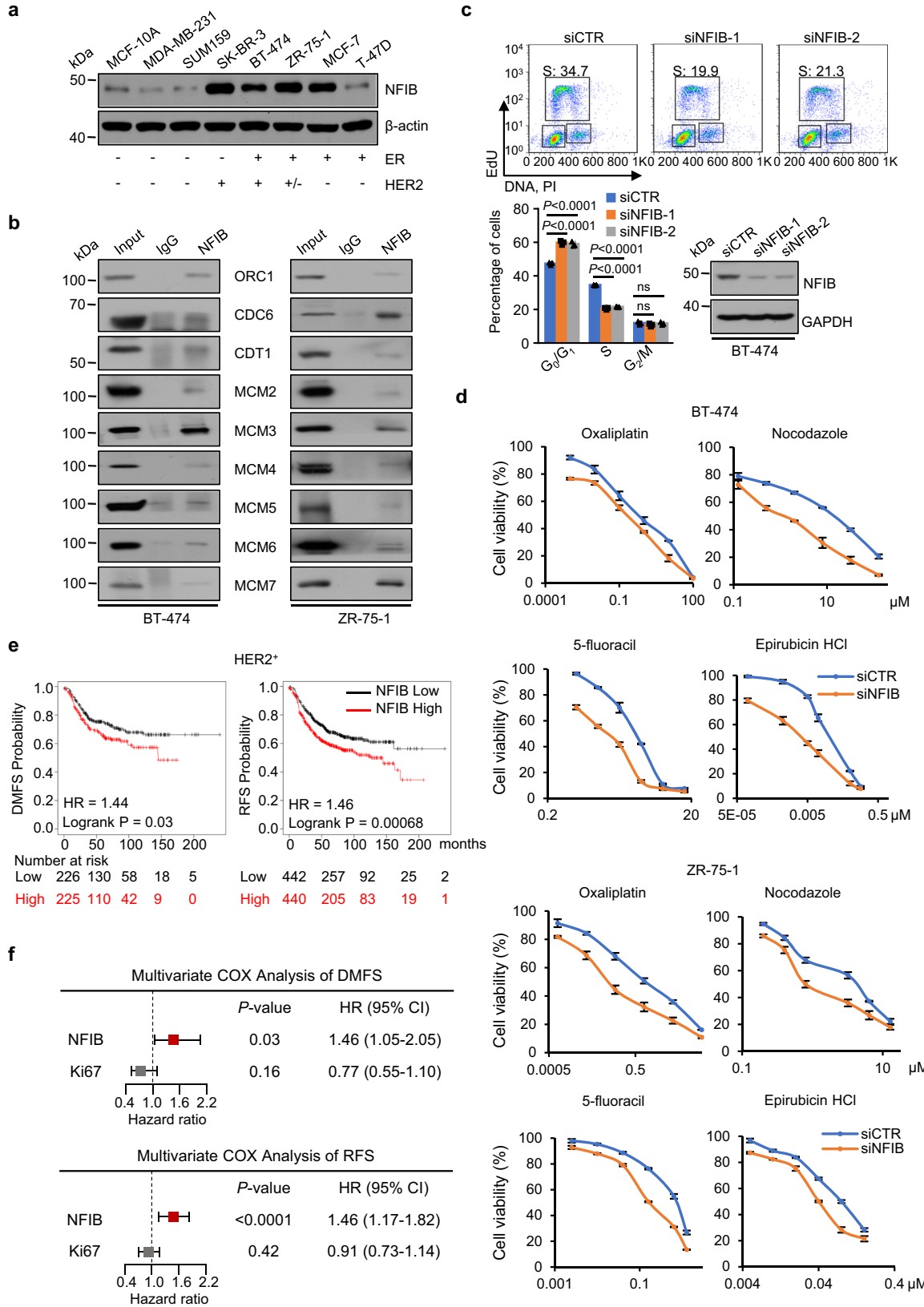

purified by glutathione-affinity resin (Amersham Biosciences). Fusion proteins (10, 15, or 20 µg) were incubated with 10 µg of native calf thymus histones (Worthington) in binding buffer (50 mM Tris-HCl, pH 7.5, 1 M NaCl, 1% Nonidet P-40, 0.5 mM EDTA, 1 mM phenylmethyl sulfonyl fluoride (PMSF) plus protease inhibitors (Roche)) at 4 °C for 4 h. Alternatively, GST fusion proteins were

incubated with 10 µg of recombinant histone octamers, which were prepared by mixing the four unfolded *Xenopus laevis* recombinant histones in equimolar amounts as previously described. Protein complexes were pulled down with glutathione beads, washed five times with the binding buffer, and subjected to western blotting using indicated antibodies.

**Fig. 8 | NFIB overexpression promotes breast cancer carcinogenesis. a** Western blotting analysis of NFIB expression in various breast cancer cell lines. The experiment was performed three times with similar observations. **b** Co-immunoprecipitation in BT-474 cells (left) or ZR-75-1 cells (right) with chromatin fraction using antibodies against NFIB followed by immunoblotting (IB) with antibodies against the indicated proteins. The experiment was performed three times with similar observations. **c** NFIB-deficient BT-474 cells were subjected to FACS analysis with EdU incorporation. The percentage of cells in S phase of the cell cycle was determined by dual PI/EdU staining. The data are presented as the means ± SD for triplicate experiments. *P* values were determined by two-way ANOVA followed by the Dunnett test. NS, not significant. The efficiency of knockdown was verified by western blotting. **d** Control or NFIB-deficient BT-474 or ZR-75-1 cells were treated with the indicated antineoplastic agents, and CCK-8 assays were performed in triplicate experiments. **e** Kaplan-Meier survival analysis for the correlation between NFIB expression and distant metastasis-free survival (DMFS) or relapse-free survival (RFS) in HER2[+] breast cancers using the online tool (http://kmplot.com/analysis/). HR, hazard ratio. **f** Multivariate COX regression with Ki-67 and NFIB of DMFS and RFS were performed in HER2[+] breast cancer patients. All bars correspond to 95% CIs.

## EM analysis of genomic DNA

In vivo, psoralen crosslinking, isolation of total genomic DNA, and enrichment of the replication intermediate and their EM visualization were performed as previously described[88]. In brief, cells were harvested, and genomic DNA was cross-linked by two rounds of incubation in 10 μM 4,5,8-trimethylpsoralen and 2 min of irradiation with 366-nm UV light. Cells were lysed, and genomic DNA was isolated from the nuclei by 100 μl proteinase K (20 mg/ml) digestion at 50 °C for 1–2 h until the solution was clear, followed by the phenol-chloroform extraction. Purified DNA was digested with Pvu II, and replication intermediates were enriched on a benzoylated naphthoylated DEAE cellulose column. EM samples were prepared by spreading the denatured DNA on carbon-coated grids and visualized by platinum rotary shadowing. Images were acquired on a microscope (JEOL JEM-1400; FEI) and analyzed with ImageJ. Daughter and parental strands were identified based on the following parameters: (a) Strand symmetry: daughter strands are likely to have the same length because the genomic DNA was digested by a sequence-specific restriction enzyme; (b) Fork structure: each DNA strand from the parental duplex continues into one of the daughters. Nucleosome density is expressed by R-value, which was calculated as the combined contour length of all nucleosome bubbles in a given stretch of DNA, divided by the overall contour length of the same DNA stretch. A reduced R-value indicates a reduction in nucleosome density.

## Protein purification with the baculovirus expression system

His$_6$-NFIB was purified in the baculovirus-driven expression system as described previously[55]. Briefly, Sf9 cells ($1.5–2 \times 10^6$/ml) were infected with baculovirus containing His$_6$-NFIB and incubated at 27 °C for 72 h. The infected cells were collected by centrifugation, washed with ice-cold PBS, and lysed in lysis buffer (150 mM NaCl, 20 mM Tris-HCl, pH 8.0, 0.05% NP-40, 10% glycerol, 1 mM PMSF). The cell extracts were incubated with Ni-NTA agarose (GE Healthcare) for 4 h at 4 °C. The resins were washed with a buffer containing 20 mM imidazole, and bound proteins were eluted with 200 mM imidazole. The fractions containing NFIB were dialyzed against BC-200 buffer (200 mM NaCl, 10 mM Tris-HCl, pH 8.0, 0.5 mM EDTA, 20% glycerol, 1 mM DTT, 1 mM PMSF) and stored at −80 °C.

## Nucleosome reconstitution

A 213-bp DNA template containing a single 601 sequence was prepared by PCR from plasmid using a biotin (bio)-labeled forward primer and a digoxigenin (dig)-labeled reverse primer. Equimolar amounts of individual histones in unfolding buffer (7 M guanidinium HCl, 20 mM Tris-HCl, pH 7.5, 10 mM DTT) were dialyzed into refolding buffer (2 M NaCl, 10 mM Tris-HCl, pH 7.5, 1 mM EDTA, 5 mM 2-mercaptoethanol), and purified through a Superdex S200 column. The reconstitution reaction mixture with histone octamers and 601 based DNA templates in TEN buffers (10 mM Tris-HCl, pH 8.0, 1 mM EDTA, 2 M NaCl) were dialyzed for 16 h at 4 °C in TEN buffer, which was continuously diluted by slowly pumping in TE buffer (10 mM Tris-HCl, pH 8.0, 1 mM EDTA) to a lower concentration of NaCl from 2 M to 0.6 M. Samples were collected after final dialysis in measurement HE buffer (10 mM HEPES, pH 8.0, 1 mM EDTA) for 4 h.

## Nucleosome binding assays

Nucleosomes for binding assays were reconstituted on 213-bp 601 DNA fragments using the salt-dialysis method as described above[56]. Different amounts of NFIB were incubated with 0.1 μg nucleosomes at 4 °C for 30 min in reaction buffer (10 mM HEPES, pH 8.0, 1 mM EDTA, 60 mM NaCl) prior to electrophoresis on 5% native PAGE electrophoresis in 0.25 × TBE buffer (22.5 mM Tris, pH 8.0, 22.5 mM boric acid, 0.5 mM EDTA) for 1 h at 80 V.

## Single-molecule magnetic tweezer experiments

The single-molecule stretching experiments were performed by magnetic tweezers[56]. The two ends of the 409-bp DNA construct were tethered via digoxigenin and anti-digoxigenin ligation to a glass coverslip and via biotin-streptavidin ligation to a 2.8 μm diameter Dynabeads (M280, Invitrogen Norway). Two small NdFeB magnets on the DNA constructs were controlled to pull on the Dynabeads and thus stretch the DNA molecule. The real time position of the bead was monitored by a CCD camera (MC1362, Mikrotron) at 200 Hz through an inverted microscope objective (UPLSAPO60XO, NA 1.35, Olympus). The extension (end-to-end distance) of the DNA construct was determined at nanometer resolution by analyzing the diffraction pattern of Dynabeads.

## ATAC-seq

About $5 \times 10^4$ U2OS cells synchronized at $G_1$/S or released into S phase were obtained as described in the results. ATAC-seq libraries were generated as described previously[89]. Peaks were called on the merged set of all ATAC-seq reads using MACS2. The differential accessible peak was assessed using DESeq2. Regions were called differentially accessible if the absolute value of the log2 fold changed (1.2) at a *p*-value < 0.01.

## CUT&Tag

Approximately $1 \times 10^5$ cells were used for each CUT&Tag assay. DNA libraries were established by NovoNGS CUT&Tag 2.0 High-Sensitivity Kit. In depth whole-genome DNA sequencing was performed by Berry Genomics (Beijing, China). The raw sequencing image data were examined by the Illumina NovaSeq analysis pipeline. Before read mapping, clean reads were obtained from the raw reads by removing the adaptor sequences. The clean reads were then aligned to the unmasked human reference genome (UCSC GRCh37, hg38) using BWA program and further analyzed by MACS2 with the false discovery rate (FDR) of 0.05 for NFIB and ORC1. The colocalization of genomic loci was investigated with the Integrative Genomics Viewer (IGV).

## Nascent strand DNA-seq

Nascent strand DNA seq was performed as previous reported[48]. Briefly, DNA fractions (0.5–2 kb) were isolated using a sucrose gradient. Half of the total purified DNA was treated with RNAase A. All DNA was then treated with lambda exonuclease (NEB, M0262S) three times and purified with QIAquick PCR Purification Kit. The ssDNA was converted to dsDNA using the Klenow and DNA Prime Labeling System (Invitrogen, 18187013). The dsDNA was then subjected to DNA library preparation using VAHTS Universal DNA Library Prep Kit (Vazyme,

ND604) according to the manufactory's instructions. The libraries were sequenced using Illumina platforms. The clean reads were then aligned to the unmasked human reference genome (UCSC GRCh37, hg38) using BWA program and further analyzed by MACS2 with the FDR of 0.05. RNase-treated samples were used to remove non-NS derived noise using Fisher's exact test. Peaks with low signals after RNase digestion remained as true NS peaks (FDR ≤ 0.05, log2 (untreated/treated) ≥ log2(1.2)).

## Micro-C

Micro-C libraries were prepared following the Dovetail™ Micro-C Kit. Briefly, for each library, $1 \times 10^5$ cells were crosslinked with 0.3 M DSG and 37% formaldehyde for 10 min at room temperature. Cells were lysed and chromatin was digested for exactly 15 min at 22 °C with MNase, proximity ligation of was performed for 5 h with chromatin capture beads to bind chromatin, end polishing, bridge ligation, and intra-aggregate ligation. DNA was isolated after reverse crosslinking at 55 °C with proteinase K. The library preparation did not require fragmentation. After end repair and adapter ligation, purified adaptor-ligated DNA was pulled down by streptavidin beads on a magnetic stand. PCR was performed using HotStart PCR Ready mix and universal PCR primer to generate the final Micro-C library. Primers were removed with SPRIselect beads. The libraries were sequenced using Illumina platforms.

Sequencing data were aligned to the GRCh38 reference genome and converted to valid pairs using dovetail-genomics/Micro-C tool (https://github.com/dovetail-genomics). The A/B compartment analysis was performed using FAN-C (0.9.0) at a 250-kb resolution. All compartments were assigned to active (A) and inactive (B) compartments based on gene density in each of the 250-kb regions. We used t-test to define the strengthened or weakened compartment with a p-value cutoff of 0.05. Loops were annotated using HiCCUPS at resolutions of 5 kb, 10 kb, and 25 kb with default Juicer parameters. Cross-analysis of Micro-C results in $G_1/S$ phase with the CUT&Tag results were performed by calculating the average number of N1 peaks within each type of 250 kb interval.

## Replication sequencing

Repli-seq assays were performed with a modified protocol of Nasent-EdU-IP seq[90]. Briefly, U2OS cells were transfected with control siRNA or NFIB siRNA and synchronized to $G_1/S$ phase by thymidine and aphidicolin double treatments. Cells were cultured with 20 μM of EdU for 10 min after release into S phase by washing out aphidicolin for 0, 4, and 6 h, respectively. The cells were collected and lysed immediately in lysis buffer (50 mM Tris-HCl, pH 8.0, 10 mM EDTA, 1 M NaCl, 0.5% SDS, 0.1 mg/ml RNase A, 0.2 mg/ml protease K), and the lysates were incubated at 37 °C for 3 h and then at 55 °C for 8 h before being sonicated for 5 cycles using Bioruptor with 30 s on and 30 s off for 30 cycles. DNA fragments ranging from 200 bp to 400 bp were selected with AMPure beads (0.6× to 1.0×) (Beckman, A63881). Biotin was conjugated to EdU-labeled DNAs by Click-iT reaction in buffer containing 15 mM Tris, pH 7.5, 400 uM biotin azide; 100 μM: 500 μM CuSO4 (Sigma, 209198):THPTA (Sigma, 762342), 5 mM freshly prepared sodium ascorbate (Sigma, A4034) at 34 °C for 20 min. DNAs were purified with MinElute PCR purification kit (Qiagen, 28004), end-repaired, dA-tailed, and adapter-ligated before being dissolved in 1× TE buffer and incubated with M280 streptavidin beads (Thermo Fisher, 11205D) in binding/washing buffer containing 5 mM Tris-HCl, pH 7.5, 1 M NaCl, 0.5 mM EDTA, pH 8.0 for 20 min. After rinsing in binding/washing buffer containing 0.5% IGEPAL CA-630 (Sigma, I3021) for 6 times, recovered DNAs from beads were used for Repli-seq library construction.

## EdU incorporation assay and flow cytometry

EdU incorporation assay was performed according to the protocol of Click-iT® EdU Flow Cytometry Assay Kits (Invitrogen). Briefly, cells were pulse-labeled with 10 μM EdU for 1 h. After harvesting and washing with 1% BSA in PBS, cells were fixed in Click-iT® fixative for 15 min at RT. Cells were dislodged in 1× Click-iT® saponin-based permeabilization and wash reagent at RT for 15 min. Click-iT® reactions were performed with Alexa Fluor® 647 azide at RT for 30 min. After digesting RNAs with RNase A and staining cells with propidium iodide, flow cytometry was performed in a FACS Calibur2 (BD), and data was analyzed with FlowJo.

## Colony formation assays

U2OS cells or MCF-10A cells were maintained in culture media in 6-well plate for 14 days, fixed with 4% paraformaldehyde, stained with 0.1% crystal violet for colony observation, and counted using a light microscope. Each experiment was performed in triplicate and repeated at least three times.

## Cesium chloride gradient centrifugation

Re-replication analysis of DNA was performed by cesium chloride gradient centrifugation assay with minor modifications[61]. Briefly, MCF-10A cells were treated with 100 mM BrdU for 14 h and lysed in lysis buffer (20 mM Tris-HCl, pH 8.0, 4 mM EDTA, 20 mM NaCl, and 2% SDS) supplemented with RNase A at 37 °C for 2 h followed by incubation with proteinase K at 55 °C for 3 h. DNA was extracted and then digested with RNase A at 37 °C overnight. About 100 μg DNA was mixed with CsCl (1 g/mL) in TE buffer (refraction index of 1.4015–1.4031). The CsCl gradient was centrifuged at 168,906 g in an NVT-65 rotor at 25 °C for 18.5 h. Fractions were collected from the bottom of the gradient in 200 μL aliquots and dialyzed against 0.1×TE buffer. DNA concentration was measured by Qubit 3.0.

## TaqMan copy number assays

For *ADORA1*, *ERBB2*, or *EXO1*, purified DNA (200 ng) from control or NFIB stably expressing MCF-10A cells was digested with 0.2 units of Nla III for 1 h at 37 °C. The restricted DNA was added to qPCR Mastermix at 20 ng per 20 μL of qPCR reaction. The ADORA1 (Hs06510210_cn), ERBB2 (Hs05480244_cn), and EXO1 (Hs01858844_cn) were purchased as 20× premix of primers and FAM-MGBNFQ probes (Applied Biosystems). All CNV assays were duplexed with an RNase P reference assay (Applied Biosystems). Thermal cycling conditions were 95 °C × 10 min (1 cycle), 94 °C × 30 s and 60 °C × 60 s (40 cycles), 98 °C × 10 min (1 cycle), and 12 °C hold.

## Drug sensitivity test

MCF-10A cells with or without NFIB overexpression or BT-474 or ZR-75-1 cells transfected with control or NFIB siRNA were seeded in 96-well plates overnight prior to the addition of antineoplastic drugs. Drug free controls were included in each assay. Plates were incubated at 37 °C for additional 3 days in a humidified atmosphere with 5% $CO_2$ followed by cell viability assessment using CCK-8 assays in triplicate experiments. $IC_{50}$ was calculated using GraphPad Prism software and P value was calculated using extra sum-of-squares F-test.

## Statistical analysis

Results were reported as mean ± SD for triplicate experiments unless otherwise noted. SPSS version 17.0, 2-tailed t test, and 1-way or 2-way ANOVA were used as indicated in the legends.

## Reporting summary

Further information on research design is available in the Nature Portfolio Reporting Summary linked to this article.

# Data availability

The raw high-throughput sequencing data generated in this study have been deposited in the NCBI Gene Expression Omnibus (GEO) database under accession code GSE201867. The source data for Figs. 1a-g, 2a, 2b,

2e, 4b-4e, 5a-5f, 5h, 6a, 6c-6i, 7d, 7e, 8a-8d and Supplementary Figs 1a, 2b-2c, 3a-3c, 5a, 5b are provided in Source Data file.

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

## Acknowledgements

This work was supported by grants (2021YFA1300603 to Y.S. and 2019YFA0508904 to J.L.) from the Ministry of Science and Technology of China, and grants (82188102, 81730079, and 31991164 to Y.S., and 81874161 and 82273155 to J.L.) from the National Natural Science Foundation of China.

## Author contributions

W.Z., J.L. and Y.S. conceived and designed the projects; W.Z. and Yue.W. designed and performed the molecular experiments; Y.L. conducted bioinformatics analyses under the instruction of Q.T.; C.L. and Yizhou.W. performed the single-molecule magnetic tweezer experiments under the instruction of P.C. and G.L.; W.Z., J.L. and Y.S wrote the manuscript; L.H., X.C., Y.P., L.X., X.W., J.W., Y.Z., L.S. analyzed the data and revised the manuscript.

## Competing interests

The authors declare no competing interests.
