## [Peer Review File · Nature Communications]

NFIB facilitates replication licensing by acting as a genome organizerEditorial Note: This manuscript has been previously reviewed at another journal that is not operating a transparent peer review scheme. This document only contains reviewer comments and rebuttal letters for versions considered at *Nature Communications*.

REVIEWER COMMENTS

Reviewer #3 (Remarks to the Author):

This submission reports an association with NFIB and members of the pre-replication complex. The paper provides biochemical evidence supporting the conclusion that NFIB, which primarily associated with chromatin during the G1 and early S phases of the cell cycle, interacted with members of the MCM complex, the replicative helicase, as well as ORC1 and CDT1, which recruit MCM to chromatin. Subsequent analyses suggest that NFIB facilitates chromatin accessibility and promotes the eviction of histones, advances assembly of the pre-replication complex on chromatin and shapes chromosome contacts. The paper also supports an association between NFIB over-expression and genetic alterations in some cancer cells.

This is a revised submission. The revision has addressed most of the previous review's concerns. Clarifying the statements and focusing on pre-RC assembly, especially pertinent to re-replication, improved clarity. Utilizing NS-seq to delineate replication origin activity is also very useful. Remaining concerns and suggestions for the current revised submission are listed below.

1. The conclusion that NFIB depletion differentially affects the activity of origins at early-replication domains still requires further substantiation. Specifically, the current submission provides evidence that twice as many origins show increased activity (fraction S2) than decreased activity (fraction S1) following NFIB depletion, and the extent of increase is lower than the extent of decrease. Yet, the paper focuses on the replication timing of fraction S1 (compared to all origins) and concludes that NFIB affects the activity of early-replicating origins, which are enriched for this fraction. The current submission does not provide replication timing analyses for fraction S2. It would be unclear, therefore, if origins with reduced activity upon NFIB depletion are normally activated at a particular time during S-phase, and if NFIB affects them in any particular way. It would be useful to include a clear analysis of the replication time of origins in fraction S2 and discuss the possible effects of NFIB in facilitating as well as suppressing origin activity.
2. Perhaps related to the above, as the paper shows an effect of NFIB on overall cell cycle progression, it would be useful to address the question of whether NFIB depletion altered the proportion of cells in early vs. mid or late S-phases, which might have indirectly altered the proportion of early, mid- and late S-phase origins in the above experiment.
3. The revision provides evidence that although ORC1 and NFIB binding sites each colocalize in "open" chromatin, ORC1 sites clearly do not delineate active replication origins; it also provides evidence that only about half the NFIB binding sites colocalize with ORC1 binding (figure 2C and the supplemental data to reviewer #3). Because the paper provides clear evidence that NFIB, directly or indirectly, facilitates CDT1 binding to chromatin, do CDT1 binding sites colocalize with NFIB binding sites?
4. Because of the significance of these data to the field, consider including data demonstrating ORC1 high colocalization with open chromatin but low colocalization with active replication origins in the supplemental portion of the paper.
5. Figure 6I: why does the NFIB sample show a shift in the location of the LL peak in the density gradient? Is this reproducible, and if not, would the HH fraction be detectable in a replica experiment in which all the other fractions align?
6. Access to all the data should be provided on a public database.

RE: Nature Communications NCOMMS-23-15621A

“NFIB Facilitates Replication Licensing by Acting as a Genome Organizer”

Response to reviewers' comments-

Reviewer #3:

Remarks to the Author:

This submission reports an association with NFIB and members of the pre-replication complex. The paper provides biochemical evidence supporting the conclusion that that NFIB, which primarily associated with chromatin during the G1 and early S phases of the cell cycle, interacted with members of the MCM complex, the replicative helicase, as well as ORC1 and CDT1, which recruit MCM to chromatin. Subsequent analyses suggest that NFIB facilitates chromatin accessibility and promotes the eviction of histones, advances assembly of the pre-replication complex on chromatin and shapes chromosome contacts. The paper also supports an association between NFIB over-expression and genetic alterations in some cancer cells.

This is a revised submission. The revision has addressed most of the previous review's concerns. Clarifying the statements and focusing on pre-RC assembly, especially pertinent to re-replication, improved clarity. Utilizing NS-seq to delineate replication origin activity is also very useful. Remaining concerns and suggestions for the current revised submission are listed below.

1. The conclusion that NFIB depletion differentially affects the activity of origins at early-replication domains still requires further substantiation. Specifically, the current submission provides evidence that twice as many origins show increased activity (fraction S2) than decreased activity (fraction S1) following NFIB depletion, and the extent of increase is lower than the extent of decrease. Yet, the paper focuses on the replication timing of fraction S1 (compared to all origins) and concludes that NFIB affects the activity of early-replicating origins, which are enriched for this fraction. The current submission does not provide replication timing analyses for fraction S2. It would unclear, therefore, if origins with reduced activity upon NFIB depletion are normally activated at a particular time during S-phase, and if NFIB affects them in any particular way. It would be useful to include a clear analysis of the replication time of origins in fraction S2 and discuss the possible effects of NFIB in facilitating as well as suppressing origin activity.

Authors: We showed in our manuscript that NFIB-facilitated the pre-RC loading mainly occurs at G₁ phase. Depletion of NFIB caused changes in replication pattern rather than decrease in overall firing, possibly due to firing compensatory origins. To comply with the reviewer's request, we have performed additional analysis of replication timing for S2 fraction. Compared

to total NS-seq peaks, S2 peaks exhibited similar pattern as S1.non-N1 peaks, which distributed throughout S phase but relatively leaned towards early-replication domains (Figure 3B). Therefore, while NFIB directly facilitates origin activity (S1.N1 peaks) at early-replication time, replication origins indirectly affected by NFIB (S1.non N1 or S2) preferentially locate at early-replication domains. This is conceivable since dormant origins fired by a compensatory mechanism are more likely close to each other. We have added the analysis and discussion in the revision.

2. Perhaps related to the above, as the paper shows an effect of NFIB on overall cell cycle progression, it would be useful to address the question of whether NFIB depletion altered the proportion of cells in early vs. mid or late S-phases, which might have indirectly altered the proportion of early, mid-and late S-phase origins in the above experiment.

Authors: To address the reviewer's concern, we have analyzed the cell cycle progression in U2OS cells with or without NFIB depletion (siCTR, siNFIB-1, siNFIB-2). The results showed that while depletion of NFIB significantly inhibited S phase progression, no significant changes were detected for the proportion of cells in early vs mid or late S-phases (Figure S3, siCTR and siNFIB-1 were used for repli-seq experiments).

3. The revision provides evidence that although ORC1 and NFIB binding sites each colocalize in "open" chromatin, ORC1 sites clearly do not delineate active replication origins; it also provides evidence that only about half the NFIB binding sites colocalize with ORC1 binding (figure 2C and the supplemental data to reviewer #3). Because the paper provides clear evidence that NFIB, directly or indirectly, facilitates CDT1 binding to chromatin, do CDT1 binding sites colocalize with NFIB binding sites?

Authors: We propose in our manuscript that NFIB acts upstream of origin licensing by opening up the chromatin to facilitate the pre-RC loading. Since ORC1 is the first pre-RC factor loaded onto the chromatin, high-throughput sequencing analysis was mainly performed with ORC1 to support our point. While CDT1 is an important component of pre-RC, this factor acts downstream of ORC1, and its chromatin distribution does not completely delineate active replication origins either due to factors such as redundant origin selection and compensatory firing of dormant origins. We thus defined active origins by NS-seq and analyzed the coexistence of NFIB and pre-RC components at two representative active origins by qChIP assays in Figure 4A-4D.

4. Because of the significance of these data to the field, consider including data demonstrating ORC1 high colocalization with open chromatin but low colocalization with active replication origins in the supplemental portion of the paper.

Authors: To respond to the reviewer, we have performed cross-analysis of ORC1 CUT&Tag

(ORC1 distribution), ATAC-seq (open chromatin, G₁ phase), and NS-seq (active replication origins) in wild-type U2OS cells. The data are presented as the Venn diagram in Figure S1C. The results showed that 87.2% of ORC1 peaks (19,997/22,915) co-localized with ATAC-seq peaks, among which 21.6% (4,320/19,997) co-localized with NS-seq-defined active origins. The overlapping ratio is consistent with the previous reports (Dellino et al, 2013; Long et al, 2020; Petryk et al, 2016).

Dellino, G.I. et al. Genome-wide mapping of human DNA-replication origins: levels of transcription at ORC1 sites regulate origin selection and replication timing. *Genome Res* 23, 1-11 (2013).

Long, H. et al. H2A.Z facilitates licensing and activation of early replication origins. *Nature* 577, 576-581 (2020).

Petryk, N. et al. Replication landscape of the human genome. *Nat Commun* 7, 10208 (2016).

5. Figure 6I: why does the NFIB sample show a shift in the location of the LL peak in the density gradient? Is this reproducible, and if not, would the HH fraction be detectable in a replica experiment in which all the other fractions align?

Authors: The shift of L:L peak in the previous density gradient assay could be a result of insufficient BrdU incorporation or incomplete digestion. We thank the reviewer for raising this point. We have optimized the assay conditions, repeated the experiments and replaced the data in the revision. The major conclusion from the experiment remains unchanged, as evidenced by the higher level of re-replicated (H:H peak) DNAs in NFIB-overexpressing cells. The shift of L:L peak now was not evident (Figure 6I).

6. Access to all the data should be provided on a public database.

Authors: All the data have been provided in the Source data profile.

REVIEWERS' COMMENTS

Reviewer #3 (Remarks to the Author):

The revision has addressed my concerns.

RE: Nature Communications NCOMMS-23-15621B
“NFIB Facilitates Replication Licensing by Acting as a Genome Organizer”

Response to reviewers' comments-

Reviewer #3:

Remarks to the Author:

The revision has addressed my concerns.

Authors: We thank the reviewer's comments.